# Evaluating NO$_X$ emissions and their effect on O$_3$ production in Texas using TROPOMI NO$_2$ and HCHO

Daniel L. Goldberg[*,1], Monica Harkey[2], Benjamin de Foy[3], Laura Judd[4], Jeremiah Johnson[5], Greg Yarwood[5], Tracey Holloway[2,6]

[1]Department of Environmental and Occupational Health, Milken Institute of Public Health, George Washington University, Washington, DC, USA
[2]Nelson Institute Center for Sustainability and the Global Environment (SAGE), University of Wisconsin-Madison, Madison, WI, USA
[3]Department of Earth and Atmospheric Sciences, Saint Louis University, St. Louis, MO, USA
[4]NASA Langley Research Center, Hampton, VA, USA
[5]Ramboll, Novato, CA, USA
[6]Department of Atmospheric & Oceanic Sciences, University of Wisconsin-Madison, Madison, WI, USA
*Correspondence to*: Daniel L. Goldberg (dgoldberg@gwu.edu)

## Abstract

The Tropospheric Monitoring Instrument (TROPOMI) on the Sentinel-5 Precursor (S5P) satellite is a valuable source of information to monitor the NO$_X$ emissions that adversely affect air quality. We conduct a series of experiments using a $4 \times 4$ km$^2$ Comprehensive Air Quality Model with Extensions (CAMx) simulation during April – September 2019 in east Texas to evaluate the multiple challenges that arise in reconciling the NO$_X$ emissions in model simulations with TROPOMI. We find an increase in NO$_2$ (+17% in urban areas) when transitioning from the TROPOMI NO$_2$ version 1.3 algorithm to the version 2.3.1 algorithm in east Texas, with the greatest difference (+25%) in the city centers and smaller differences (+5%) in less polluted areas. We find that lightning NO$_X$ emissions in the model simulation contribute up to 24% of the column NO$_2$ in the areas over the Gulf of Mexico and 8% in Texas urban areas. NO$_X$ emissions inventories, when using locally resolved inputs, agree with NO$_X$ emissions derived from TROPOMI NO$_2$ version 2.3.1 to within 20% in most circumstances, with a small NO$_X$ underestimate in Dallas-Fort Worth (– 13%) and Houston (– 20%). In the vicinity of large power plant plumes (e.g., Martin Lake and Limestone) we find larger disagreements: the satellite NO$_2$ is consistently smaller by 40 – 60% than the modelled NO$_2$, which incorporates measured stack emissions. We find that TROPOMI is having difficulty distinguishing NO$_2$ attributed to power plants from the background NO$_2$ concentrations in Texas – an area with atmospheric conditions that cause short NO$_2$ lifetimes. Secondarily, the NO$_X$/NO$_2$ ratio in the model may be underestimated due to the 4 km grid cell size. To understand ozone formation regimes in the area, we combine NO$_2$ column information with HCHO column information. We find modest low biases in the model relative to TROPOMI HCHO: – 9% underestimate in eastern Texas and – 21% in areas of central Texas with lower biogenic VOC emissions. Ozone formation regimes at the time of the early afternoon overpass are NOx-limited almost everywhere in the domain except along the Houston ship channel, near the Dallas Fort Worth International airport, and in the presence of undiluted power plant plumes. There are likely NOx-saturated ozone formation conditions in the early morning hours that TROPOMI cannot observe, and would be well-suited for analysis with NO$_2$ and HCHO from the upcoming TEMPO mission. This study highlights that TROPOMI measurements offer a valuable means to validate emissions inventories and ozone formation regimes, with important limitations.

## 1 Introduction

Nitrogen oxides ($NO_X \equiv NO+NO_2$) are a group of reactive trace gases toxic to human health (Burnett et al., 2004; He et al., 2020; Khreis et al., 2017) that can be converted into other chemical species, including ozone and fine particulate matter (Jacob, 1999). There are some natural emissions of $NO_X$ (e.g., lightning, soil), but the majority of the $NO_X$ emissions are from anthropogenic sources (Van Vuuren et al., 2011). Anthropogenic $NO_X$ emissions in polluted areas can be estimated using $NO_2$ column measurements from satellites (Lamsal et al., 2011; Leue et al., 2001; Martin, 2003; Stavrakou et al., 2008) if the meteorology, $NO_2$ chemical lifetime, tropospheric/stratospheric components, and $NO_X/NO_2$ ratio are all properly accounted for (Beirle et al., 2011; de Foy et al., 2014; Goldberg et al., 2020).

Satellite instruments can observe $NO_2$ from space because it has strong absorption features within the 400 – 465 nm wavelength region (Vandaele et al., 1998). By comparing observed spectra with a reference spectrum, the amount of $NO_2$ in the atmosphere between the instrument and the surface can be derived; this technique is called differential optical absorption spectroscopy (DOAS) (Platt, 1994). The first satellite instrument to utilize the DOAS technique to observe $NO_2$ air pollution was Global Ozone Monitoring Experiment (GOME) (Burrows et al., 1999) launched in 1995 ($320 \times 40$ km$^2$ spatial resolution) and was followed by the Ozone Monitoring Instrument (OMI) (Levelt et al., 2006) launched in 2004 with vastly improved pixel resolution ($24 \times 13$ km$^2$ at nadir) and instrument stability (Schenkeveld et al., 2017). Initial studies used OMI $NO_2$ satellite data to pinpoint $NO_X$ emissions in the vicinity of large power plants (Duncan et al., 2013; Kim et al., 2009; Russell et al., 2012) and in areas with high population densities (Boersma et al., 2008; Lamsal et al., 2008, 2010).

TROPOMI (Veefkind et al., 2012) builds upon the overwhelming success of OMI (Levelt et al., 2018) and has pixel resolution and instrument stability that are even more advantageous for observing urban scale $NO_2$ pollution. Most recently, TROPOMI has been used to estimate $NO_X$ emissions (Beirle et al., 2019; Dix et al., 2022; de Foy and Schauer, 2022; Goldberg et al., 2019b; Griffin et al., 2019; Lorente et al., 2019) and its changes during the COVID-19 lockdowns (Bauwens et al., 2020; Cooper et al., 2022; Goldberg et al., 2020; Liu et al., 2020; Souri et al., 2021; Sun et al., 2021; Wang et al., 2020). The high spatial resolution of TROPOMI makes it an excellent instrument to observe some of the fine-scale structure of $NO_2$ pollution, such as within cities (Demetillo et al., 2020; Geddes et al., 2021; Goldberg et al., 2021; Ialongo et al., 2020; Zhao et al., 2020), near power plants (Saw et al., 2021; Shikwambana et al., 2020), near ships (Georgoulias et al., 2020), in the presence of wildfires (Griffin et al., 2021; Jin et al., 2021), and in the presence of oil and gas operations (van der A et al., 2020; Dix et al., 2022; Ialongo et al., 2021).

Studies in the mid 2010s (Canty et al., 2015; Curier et al., 2014; Harkey et al., 2015; Kemball-Cook et al., 2015; Souri et al., 2016; Travis et al., 2016) described the synergistic use of satellite $NO_2$ and regional chemical transport model simulations to better quantify $NO_X$ emissions. These studies compared satellite data to model simulations directly while also accounting for vertical sensitivity differences between the satellite and model simulation. Results from these studies were mixed, but generally found that satellite $NO_2$ was larger than the model data in rural areas and smaller than the model in urban areas. These studies suggested a potential overestimate of $NO_X$ emissions in U.S.

urban areas, and demonstrated the importance of stratospheric transport, lightning $NO_X$ emissions, soil $NO_X$ emissions, and $NO_2$ chemical recycling.

For simulations of 2018 and more recent years, TROPOMI data have been used for model evaluations (e.g., Community Multiscale Air Quality (CMAQ) modeling system, Long Term Ozone Simulation European Operational Smog (LOTOS-EUROS) model, Weather Research Forecast with Chemistry (WRF-Chem) model). Most studies show high correlations, but larger $NO_2$ columns in the model in major urban areas and near large point sources. This result is persistent across regions including Korea (Kim et al., 2020), Europe (Skoulidou et al., 2021), and North America (Lawal et al., 2021; Li et al., 2021). Judd et al. (2020) examined $NO_2$ in New York City using TROPOMI version 1.3 (v1.3) $NO_2$ data and aircraft/ground-based spectrometer measurements and found that the satellite underestimated $NO_2$ by 19-33%. Verhoelst et al. (2021) also found a satellite low bias (23 – 51%) in v1.3 when comparing to ground-based measurements suggesting an algorithm change is a necessary.

There appears to be three primary causes for the low bias in the v1.3 algorithm: 1.) a persistent high bias of the cloud pressure retrieved with the Fast Retrieval Scheme for Clouds from the Oxygen A band (FRESCO) cloud algorithm (van Geffen et al., 2021), 2.) the relatively coarse model *a priori* vertical $NO_2$ profiles ($1° \times 1°$) which underestimate the near-surface $NO_2$ in polluted regions and are needed for the conversion of the satellite slant column into a vertical column (Goldberg et al., 2017), and 3.) the spatial heterogeneity in pointwise-to-gridded data comparisons (Souri et al., 2022). The TROPOMI version 2.3.1 (v2.3.1) $NO_2$ algorithm includes an improved way to estimate cloud pressure and addresses reason #1. Reason #2 can be remediated by incorporating high-resolution spatial information. Judd et al. (2021) reported that when information from higher resolution chemical transport models were included in the calculation of the air mass factor, TROPOMI $NO_2$ values increased by approximately 12 – 14% in an urban area. Reason #3 can be accounted for by comparing the satellite measurements to model simulations at similar spatial resolutions as the satellite.

We conduct a series of experiments using a high-resolution photochemical grid model simulation over east Texas and evaluate multiple challenges that arise in evaluation with TROPOMI. We examine the impact of the revised TROPOMI algorithm (Section 3.1), the impact of lightning emissions and other sources of $NO_2$ in the free troposphere (Section 3.2), accounting for TROPOMI's vertical sensitivity (Section 3.3), and evaluating the ability of TROPOMI to resolve urban areas and power plants (Section 3.4). While each of these issues involves disparate aspects of model methodology and chemistry, in fact they are intertwined in the correct interpretation of satellite and model results. Based on these results, we consider the ability of TROPOMI to inform emission quantification (Section 4.1) and evaluate ozone sensitivity along with formaldehyde (HCHO) retrievals (Section 4.2). Based on these results, we offer best practice recommendations for TROPOMI model evaluation and future work.

## 2 Methods

### 2.1 CAMx model simulation

For our analysis, we use a $4 \times 4$ km$^2$ Comprehensive Air quality Model with extensions (CAMx) simulation version 7.00 centered over eastern Texas driven off-line by Weather Research Forecast (WRF) model version 4.0.3. The $4 \times 4$ km$^2$ domain is nested inside $12 \times 12$ km$^2$ and $36 \times 36$ km$^2$ two-way domains, shown in Figure 1. We ran the WRF and CAMx models for the 2019 Texas ozone season, March 15 – October 15. Only model output between April 1 through Sept 30 are used for this study. We use the $0.25° \times 0.25°$ Global Forecasting System data assimilation system as initial conditions for the WRF meteorological model, which is also used for boundary conditions and analysis nudging on the 36 km and 12 km domains. The WRF simulation had 43 vertical levels between the surface and 50 hPa, with approximately 21 layers below 700 hPa. The 43 WRF vertical layers were mapped to 28 vertical layers for the CAMx model simulations; all 21 layers below 700 hPa were mapped without merging. The CAMx simulation was utilized with the Carbon Bond Version 6, Revision 4 (CB6r4) chemical mechanism (Emery et al., 2016).

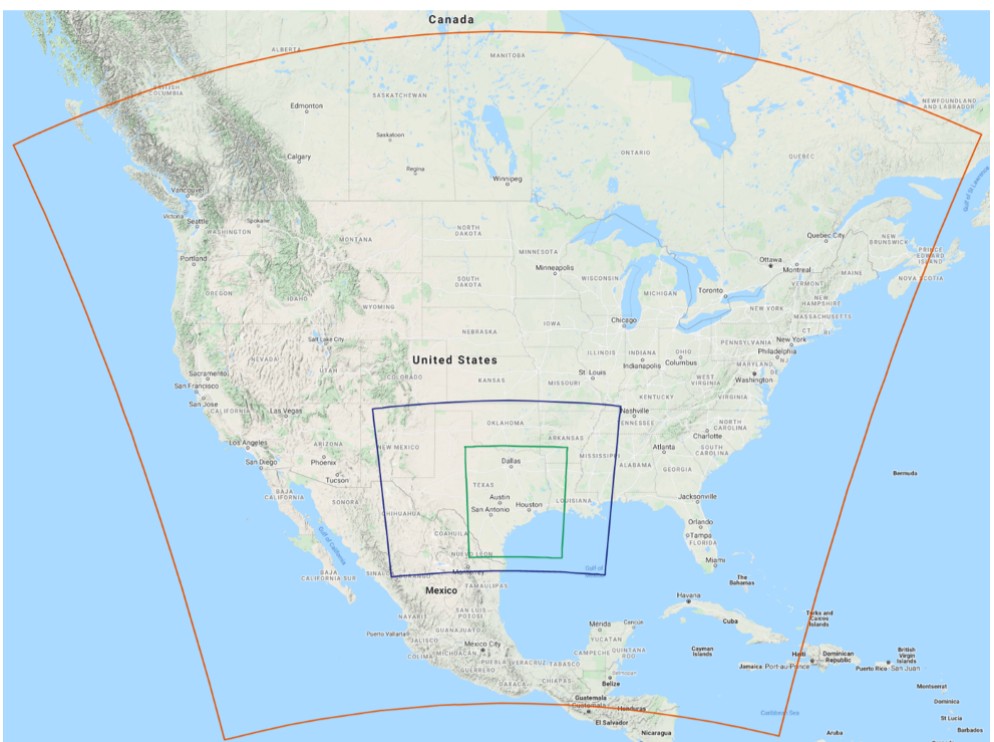

**Figure 1.** CAMx 36/12/4 km modeling domains. Image underlaid is from © Google Maps.

For this study, we use a projected 2020 Texas Commission on Environmental Quality (TCEQ) modeling inventory from a 2017 TCEQ inventory, which is different from the National Emission Inventory (NEI). The 2020 modeling emissions inventory did not include impacts of the socioeconomic response to COVID-19, which was advantageous for this application since we modelled the 2019 ozone season. TCEQ developed the 2020 modeling emissions inventory for the Dallas-Fort Worth and Houston-Galveston-Brazoria Attainment Demonstration State

Implementation Plan revision (Johnson et al., 2018). Within Texas, emissions were calculated using locally resolved inputs, such as mobile emissions from MOVES2014a adjusted based on traffic statistics from the Highway Performance Monitoring System. Outside of Texas, NEI estimates were used such as the default outputs from MOVES2014 and the 2014 EPA NEI.

5  We included hourly-specific power plant emissions using measurements from the EPA's Clean Air Markets Division (CAMD) (https://www.epa.gov/airmarkets) as inputs into the model simulation. Large power plants use Continuous Emissions Monitoring Systems (CEMS) to report emissions of sulfur dioxide ($SO_2$), NOx, and $CO_2$, along with other parameters such as heat input, as required by the federal Clean Air Act. We downloaded hourly data from EPA's Air Markets Program Data (AMPD) website for the continental US for March through October
10  2019. Stack parameters were based on EPA's 2017 NEI data. The 2017 NEI data with matching facilities in Texas were then adjusted to their 2019 annual totals. Table 1 provides the annual inventory NOx emission rates for four cities within a 50 km radius of the city center and three power plants examined in detail in this study.

**Table 1.** $NO_X$ emission rates for 2019 from the four largest metropolitan areas and three largest power plants within our model domain. For the cities, the fraction of emissions allocated to on-road mobile sources are also noted.

| Location | NOx emissions (Gg/yr) | Fraction on-road mobile sources |
|---|---|---|
| Dallas-Fort Worth (city) | 58 | 0.34 |
| Houston (city) | 86 | 0.24 |
| San Antonio (city) | 35 | 0.24 |
| Austin (city) | 23 | 0.27 |
| Martin Lake (power plant) | 8.4 | N/A |
| Limestone (power plant) | 7.1 | N/A |
| Sam Seymour (power plant) | 5.8 | N/A |

Biogenic emissions were estimated for 2019 from the Model of Emissions of Gases and Aerosols from Nature (MEGAN) version 3.1 and fire emissions from Fire INventory of NCAR (FINN) version 1. We included lightning NOx ($LNO_X$) emissions with the CAMx $LNO_X$ processor using the 2019 WRF meteorological data. The $LNO_X$ processor estimates hourly, grid column-specific lightning flash rates (Luo et al., 2017; Price and Rind, 1992) using
20  cloud top heights and Convective Available Potential Energy (CAPE) diagnosed from WRF temperature and moisture profiles. The processor then determines the ratio of intracloud lightning (IC) to cloud-to-ground lightning (CG) according to the approach of Price and Rind (1993), NO yield per flash estimated by Pickering et al. (2017), and vertical distribution of resulting NO emission rates following DeCaria et al. (2005). In-line inorganic iodine emissions ($I_X$) from saltwater areas and iodine chemistry are also included.

**2.2 TROPOMI**

TROPOMI was launched by the European Space Agency (ESA) for the European Union's Copernicus S5P satellite mission on 13 October 2017. The satellite follows a sun-synchronous, low-earth (825 km) orbit with an equator overpass time of approximately 13:30 local solar time. TROPOMI measures total column amounts of several trace gases in the Ultraviolet-Visible-Near Infrared (UV-VIS-NIR) (e.g., $NO_2$ and HCHO) and Shortwave Infrared (SWIR) (e.g., CO) spectral regions. At nadir, pixel sizes are $3.5 \times 7$ km$^2$ (modified to $3.5 \times 5.5$ km$^2$ on August 6, 2019) with the edges having slightly larger pixels sizes (~14 km wide) across a 2600 km swath, equating to 450 rows (van Geffen et al., 2020). The instrument observes the swath approximately once every second and orbits the Earth in about 100 minutes, resulting in daily global coverage. For the domain-wide comparisons, we screened TROPOMI pixels for quality assurance flag values greater than 0.75. As a polar-orbiting satellite with an afternoon overpass, care must be taken in the interpretation of TROPOMI column retrievals as an indicator of near-surface emissions (Penn and Holloway, 2020; Streets et al., 2013). TROPOMI provides "snapshots" at the same time each day, except as limited by cloud cover, surface albedo, or instrument errors.

**2.2.1 NO₂**

$NO_2$ slant column densities are derived from radiance measurements in the $405 - 465$ nm spectral window of the UV-VIS-NIR spectrometer. Tropospheric vertical column density data, which represent the vertically integrated number of $NO_2$ molecules per unit area between the surface and the tropopause, are then calculated by subtracting the stratospheric portion and then converting the tropospheric slant column to a vertical column using an air mass factor (AMF). The AMF is a unitless quantity used to convert the slant column into a vertical column and is a function of the satellite viewing angles, solar angles, the effective cloud radiance fraction and pressure, the vertical profile shape of $NO_2$ provided by a chemical transport model simulation (for operational data the TM5-MP model is used at $1 \times 1°$ resolution) (Williams et al., 2017), and the surface reflectivity (for operational data, climatological Lambertian Equivalent Reflectivity is used at a $0.5 \times 0.5°$ resolution) (Kleipool et al., 2008). The operational AMF calculation does not explicitly account for aerosol absorption effects, which are accounted for in the effective cloud radiance fraction.

For our analysis we use both the v1.3 off-line (OFFL) algorithm, which was operational during the April through September 2019 timeframe, and the v2.3.1 Products Algorithm Laboratory (PAL) algorithm, released in December 2021 and includes updates to the cloud retrieval scheme (decrease in cloud pressure), the surface albedo (to avoid negative cloud fractions), and the quality flags (better screening of snow). The net result of the change in tropospheric vertical column $NO_2$ from v1.3 to v2.3.1 has been reported to be a +13% increase for cloud-free scenes that varies spatially, and is higher in polluted areas (van Geffen et al., 2021).

**2.2.2 HCHO**

HCHO slant column densities are derived from radiance measurements in the $328 - 359$ nm spectral window of the UV-VIS-NIR spectrometer. In a similar manner to $NO_2$, HCHO is measured as a slant column – representing the

amount of HCHO between the surface and detector – and is converted from a slant column to a vertical column using an AMF and *a priori* information from TM5-MP. However, in contrast to NO$_2$, HCHO is reported only as a tropospheric vertical column amount since the stratospheric portion is negligible.

For our analysis, we use the v1.1.6 off-line (OFFL) algorithm, which was operational during the April through September 2019 timeframe. At the time of this study, there has not been a public release of TROPOMI HCHO data using the version 2 algorithm predating July 13, 2020.

### 2.2.3 Re-gridding and accounting for the vertical sensitivity of TROPOMI

For comparison with CAMx, we gridded TROPOMI data to the model to create a custom "Level-3" data product for comparison with each other or model data on a common grid. Though our Level-3 data product is on an equivalent horizontal grid as the model, the satellite *a priori* (used in the retrieval) and CAMx have different vertical resolutions and distributions of NO$_2$. To limit artificial differences when doing the comparisons in this work, additional processing is done two ways.

1.  Applying the Averaging Kernel: The most user-friendly approach involves creating a model simulated satellite NO$_2$ column using the CAMx profile and a TROPOMI data product-specific "averaging kernel," which may be described as the weights used to calculate a weighted vertical integral (we refer to this as AK). To apply the averaging kernel to the model simulation, we first interpolate the averaging kernel from the TM5-MP vertical pressure levels to the CAMx vertical pressure levels at each horizontal grid location using linear interpolation. Once the averaging kernel is on the CAMx grid, we multiply the partial tropospheric columns by the averaging kernel at each vertical level (e.g., multiply the partial columns by ~1.5 at 10 km, by ~1 at 2 km, and by ~0.5 near the surface) to account for the retrieval sensitivity at different altitudes. We applied the gridded TROPOMI NO$_2$ averaging kernel in a similar manner to previous work (Deeter, 2002; Harkey et al., 2015, 2020).

2.  Re-calculating the AMF: In a second approach, we instead use daily partial vertical NO$_2$ columns from CAMx and the tropospheric averaging kernel to recalculate a new TROPOMI AMF as described in the TROPOMI NO$_2$ Product User's Manual (Eskes et al., 2021). The tropospheric slant column is then divided by the recalculated AMF to generate day-specific recalculated tropospheric vertical column NO$_2$ (Goldberg et al., 2017; Judd et al., 2020). This new satellite measurement can then be compared directly to the tropospheric vertical column NO$_2$ from the CAMx model simulation.

### 2.3 Deriving NO$_X$ emissions from TROPOMI NO$_2$

### 2.3.1 Exponentially modified Gaussian fitting method

To derive NO$_X$ emissions from the polluted areas of east Texas, an exponentially modified Gaussian (EMG) function is fit to a collection of NO$_2$ plumes observed from TROPOMI. The original methodology, proposed by Beirle et al. (2011), involves the fitting of satellite line densities to an EMG function. Line densities are the integral of the column

NO₂ retrieval perpendicular to the path of the plume; the units are mass per distance. We rotate each day's plume based on the wind direction, so that all daily plumes are artificially in the same horizontal direction (Lu et al., 2015; Valin et al., 2013). The 100-m wind speed and direction are obtained from the ERA5 re-analysis project (Hersbach et al., 2020). Appendix B has a sensitivity analysis of using different wind configurations. Once all daily plumes are rotated and aggregated together, the EMG statistical fit can be applied as expressed as Equation (1):

$$OMI\ NO_2\ Line\ Density = \alpha \left[ \frac{1}{x_o} exp \left( \frac{\mu}{x_o} + \frac{\sigma^2}{2x_o^2} - \frac{x}{x_o} \right) \Phi \left( \frac{x-\mu}{\sigma} - \frac{\sigma}{x_o} \right) \right] + \beta \qquad (1)$$

where $\alpha$ is the total number of NO₂ molecules observed near the pollution source, excluding the effect of background NO₂, $\beta$; $x_o$ is the e-folding distance downwind, representing the length scale of the NO₂ decay; $\mu$ is the location of the apparent source relative to the assumed pollution source center; $\sigma$ is the standard deviation of the Gaussian function, representing the Gaussian smoothing length scale; $\Phi$ is the Gaussian cumulative distribution function. Using the 'curvefit' function in IDL, we determine the five unknown parameters: $\alpha,\ x_o,\ \sigma,\ \mu,\ \beta$ based on the independent (distance; x) and dependent (NO₂ column line density) variables.

Using the mean ERA5 100-m wind speed, $w$, the mean effective NO₂ lifetime $\tau_{effective}$ and the mean NOₓ emissions can be calculated from the fitted parameters $x_o$ and $\alpha$, as expressed in Equation 2:

$$NO_x\ Emissions = 1.32 \left( \frac{\alpha}{\tau_{effective}} \right), \text{ where } \tau_{effective} = \frac{x_o}{w} \qquad (2)$$

Equation 2 yields emission estimates in units of mol-s⁻¹. A factor of 1.32 is the mean column-averaged NOₓ / NO₂ ratio and is the widely used value to convert the NO₂ to NOₓ in polluted regions (Beirle et al., 2021). Appendix C shows the variation in the NOₓ / NO₂ across our domain.

**2.3.2 Flux divergence method**

Emissions were also estimated using the flux divergence method (Beirle et al., 2019) :

$$NO_x\ Emissions = 1.32 \left( \nabla \cdot (VCD \cdot \text{u}) + \frac{VCD}{\tau} \right) \qquad (3)$$

Fluxes of NO₂ were obtained by multiplying NO₂ vertical column densities (VCDs) with wind speeds (u) in orthogonal directions (along and across the swath tracks). The divergence of the fluxes yields an emission estimate in units of mol-m⁻² s⁻¹. The fluxes can then be multiplied by the urban area to get emission rates in analogous units as Equation 2. Sinks of NO₂ are included in the equation by adding VCD divided by the atmospheric lifetime of NO₂, $\tau$, which was taken from the EMG fit. Estimates of NOx emissions are obtained by multiplying the estimates by the ratio of NOx to NO₂, which is the same 1.32 value as the EMG method (Beirle et al., 2021). The fluxes were calculated using the same 100-m ERA5 wind product used for the EMG estimates. The winds were linearly interpolated to the daily swath grid. This method follows de Foy and Schauer (2022) with minor modifications. The quality assurance flag threshold was set to 0.75 to be consistent with EMG. The central 250 pixels (out of 450) were used for swaths as these

have a higher resolution than the outer bands and is critical for this method. We retrieved swaths from October 2019 through September 2021. Although this period does include the COVID-19 lockdowns, the October 2019 through September 2021 timeframe does not show time-averaged $NO_2$ values more than 10% different than the year prior, and is well within the uncertainty of this analysis. Two-dimensional Gaussian fits were obtained using the method described in de Foy et al. (2014).

## 3 Results and Discussion

### 3.1 Comparison between TROPOMI version 1.3 and version 2.3.1 algorithms

To elucidate the effects of the recent TROPOMI $NO_2$ algorithm change from v1.3 to v2.3.1, we compare both within our model domain. As expected, the v2.3.1 algorithm yields consistently larger values than the v1.3 algorithm in most areas of our east Texas domain (Figure 2). The largest increases by both magnitude and percentage occur in the most polluted areas. We find an average increase of +16.6% in urban counties, with a maximum increase of +45% in the most polluted section of east Houston. Increases exceeding +20% also occur in the vicinity of large point source emissions. In the rural areas of east Texas, we generally observe small increases less than +5%. We fit a linear regression to a scatterplot of the tropospheric vertical columns from both algorithms in the urban counties, and find a slope of 1.30 and a negative intercept, which further confirms that the algorithm change affects the most polluted areas more strongly than the moderate and low polluted areas.

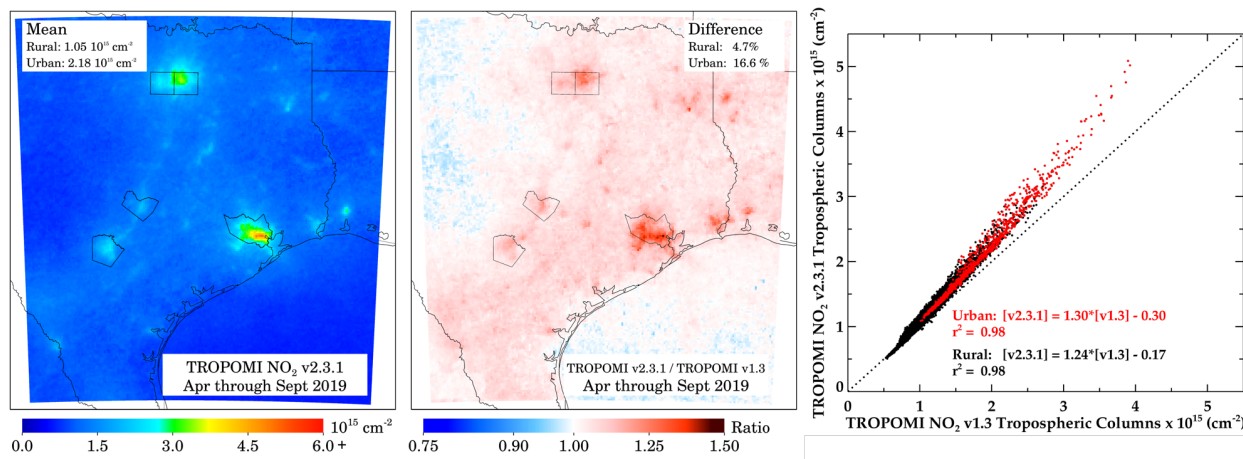

**Figure 2.** (Left) $NO_2$ tropospheric vertical column amounts from the TROPOMI $NO_2$ v2.3.1 algorithm screened with a quality assurance flag greater than 0.75. (Center) The ratio between the $NO_2$ tropospheric vertical column amounts from the v2.3.1 algorithm compared to the v1.3 algorithm. (Right) A scatterplot and linear fit between the two TROPOMI $NO_2$ products used in panel b. Urban area is defined as the five counties surrounding the largest five cities (Houston, Dallas, Fort Worth, San Antonio, and Austin). Rural area is everywhere outside those counties.

**3.2 Effects of free tropospheric NO₂ and lightning NOx**

For this study, we conducted two CAMx simulations: with and without lightning $NO_X$ emissions. The tropospheric $NO_2$ vertical profiles for eastern Texas, Dallas, and Houston are shown in the left side panels of Figure 3. In a CAMx simulation without lightning $NO_X$, average $NO_2$ concentrations between 2.5 – 10 km averaged 20 ppt for the eastern
Texas domain. This can be compared to free tropospheric (>2.5 km) $NO_2$ concentrations from the NASA Studies of Emissions, Atmospheric Composition, Clouds and Climate Coupling by Regional Surveys (SEAC4RS) campaign within our east Texas model domain, but in 2013 instead of 2019. Measured $NO_2$ concentrations between 2.5 – 10 km averaged 50 ppt during the SEAC4RS campaign. This also compares the ~40 ppt estimate from OMI using a cloud-slicing methodology in the central US during June – August 2005 – 2007 (Marais et al., 2018). When lightning $NO_X$
emissions are included in CAMx, the free tropospheric $NO_2$ between 2.5 – 10 km increases from 20 ppt to 33 ppt, but there is still a slight underestimate compared to SEAC4RS data between 2.5 – 6 km. The small underestimate shown in the CAMx simulation with lightning $NO_X$ emissions compared to the SEAC4RS data in the 2.5 – 6 km altitude range could be due the decrease in anthropogenic $NO_X$ emissions between 2013 and 2019. Collocated vertical $NO_2$ measurements in time and space would be needed to evaluate this further.

In order to compare model simulation output to satellite data, it is important to understand free tropospheric $NO_2$ (Marais et al., 2018, 2021) and understand its effects on the satellite retrieval (Silvern et al., 2019). TROPOMI has greater sensitivity to the upper portion of the troposphere and this must be accounted for in any comparison with model output. In the right panels of Figure 3, we show the modeled shape profiles – the $NO_2$ vertical distribution normalized to a unitless quantity that integrates to unity over the depth of the troposphere – and the sensitivity of TROPOMI to
$NO_2$ at different levels of the atmosphere (green line). In Texas during summer 2019, TROPOMI was three times as sensitive to $NO_2$ at an altitude of 10 km (tropospheric averaging kernel = 1.5) as compared to the surface (tropospheric averaging kernel = 0.5). This demonstrates that $NO_2$ at the tropospheric/stratospheric interface (~12 km altitude), such as lightning $NO_X$ (Zhu et al., 2019) and cruising aircraft emissions, can have an outsized effect on the satellite measurement. To facilitate a comparison, model simulated column amounts can be adjusted by "applying the
averaging kernel", which will be discussed in Section 3.3.

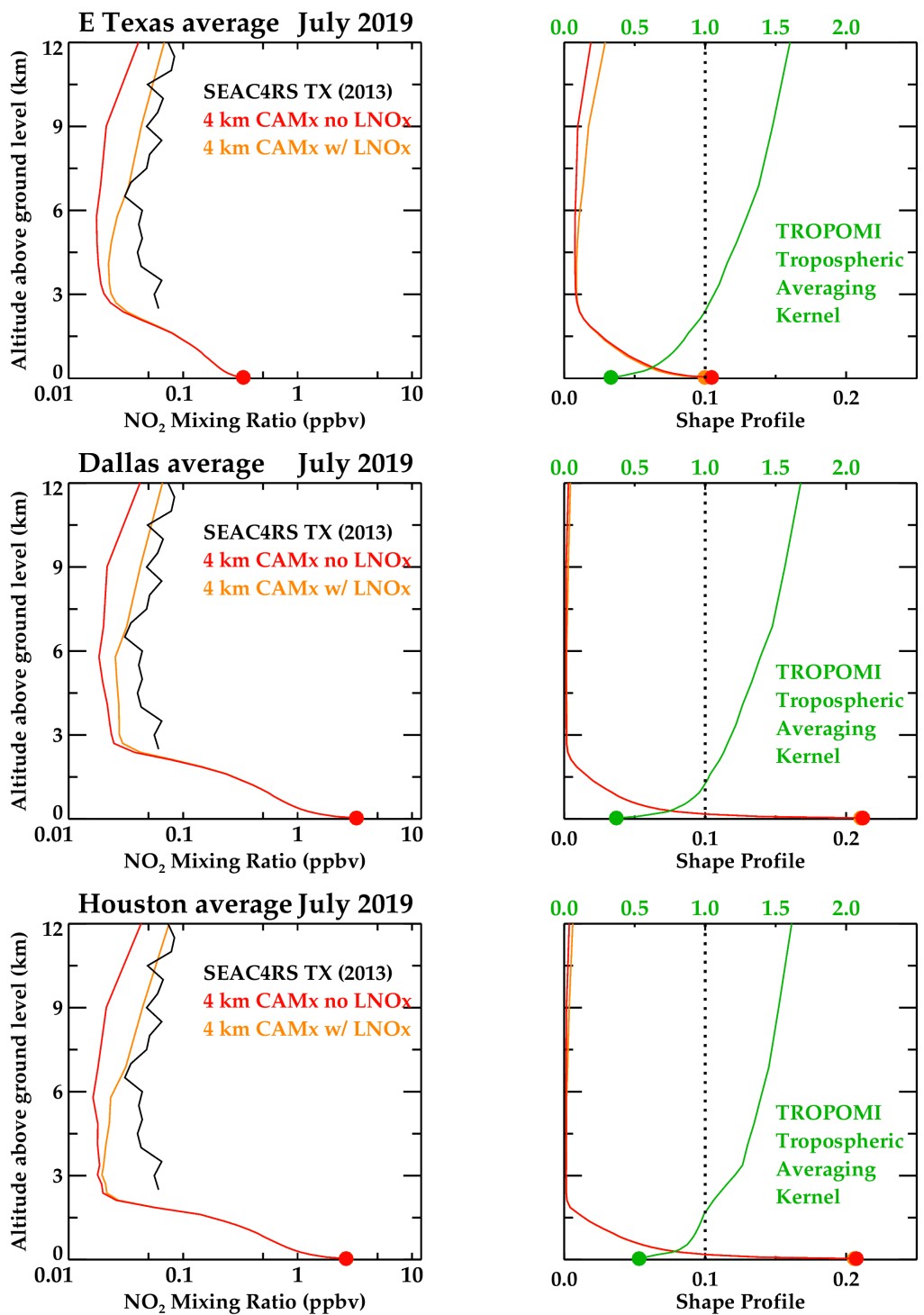

**Figure 3.** (Left) $NO_2$ vertical concentration profiles between the surface and 12 km altitude from the CAMx model with (orange) and without (red) lightning NOx emissions for July 2019, and median free tropospheric $NO_2$ in situ observations acquired during the Aug – Sept 2013 NASA SEAC4RS field campaign (black) for (top) the East Texas average, (middle) Dallas, and (bottom) Houston; orange and red dots represent surface concentrations. (Right) $NO_2$ shape profiles – the fraction of the column at any given altitude – from the same two model simulations and the TROPOMI tropospheric averaging kernel for the same locations; dotted line indicates an averaging kernel=1.

The inclusion of lightning $NO_X$ emissions increases seasonal column tropospheric $NO_2$ by an average of $0.16 \times 10^{15}$ molecules-cm$^{-2}$ in the model simulation during April through September 2019 (Figure 4). This increase varies spatiotemporally due to the prevalence of thunderstorms, however when averaged over 6 months, the increase is relatively homogeneous. The inclusion of lightning $NO_X$ emissions most affects the satellite-model comparison in rural areas, but is also relevant in urban areas. The $0.16 \times 10^{15}$ molecules-cm$^{-2}$ increase yields an increase in the tropospheric column $NO_2$ of +7.8% in urban areas, +15% in the rural areas of eastern Texas, and up to +24% over the Gulf of Mexico. For the rest of this paper, only the CAMx simulation with the inclusion of lightning $NO_X$ emissions will be analyzed.

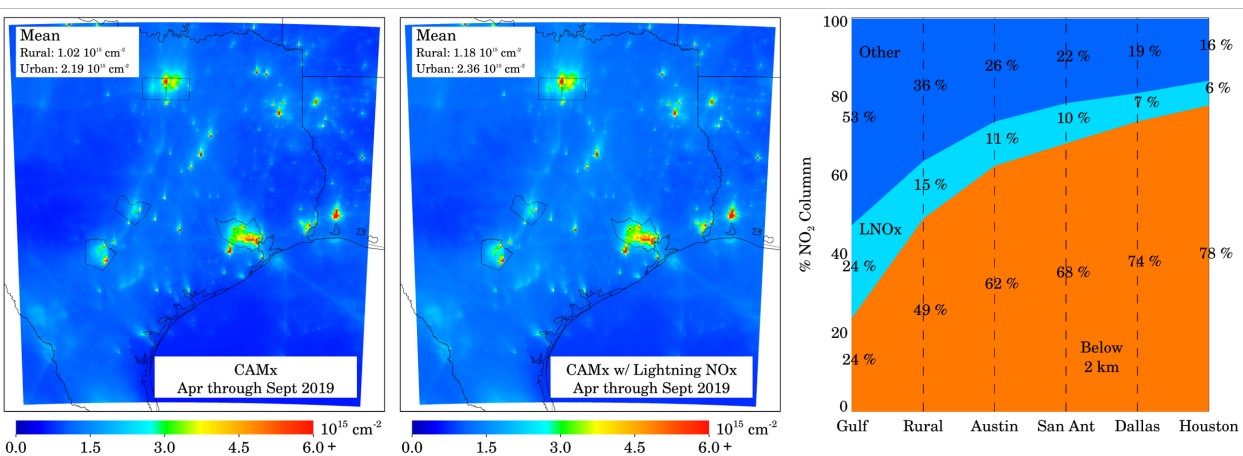

**Figure 4.** $NO_2$ tropospheric vertical column amounts from the CAMx model (left) without and (center) with lightning NOx emissions averaged during April through Sept 2019 at the coincident TROPOMI overpass time (~19 UTC). Areas with invalid TROPOMI data are similarly screened out from the model output on a daily basis. Urban area is defined as the five counties surrounding the largest five cities (Houston, Dallas, Fort Worth, San Antonio, and Austin). Rural area is everywhere outside those counties. (Right) The fraction of the $NO_2$ column attributed to different layers of the atmosphere (below 2 km, above 2 km (attributed to Other), and above 2 km attributed to lightning $NO_X$ (LNOx)) at six locations (Gulf of Mexico, rural central Texas, Austin, San Antonio, Dallas and Houston); the fraction attributed to lightning $NO_X$ (LNOx) is calculated as the $NO_2$ addition between the two simulations without and with lightning $NO_X$ emissions.

**3.3 Applying the averaging kernel and re-calculating the air mass factor**

To compare a chemical transport model simulation to satellite data, one must account for the differing assumptions about the vertical $NO_2$ distributions between model and satellite. One can either apply the averaging kernel from the satellite instrument to the $NO_2$ column from the model simulation or use the $NO_2$ vertical profile from the model simulation and the averaging kernel to re-calculate AMF and tropospheric $NO_2$ vertical column of the satellite measurement. Typically studies either use one of the two methods; similar to Douros et al. (2022) we use both.

The comparison between the model and model with the tropospheric averaging kernel (AK) applied is shown in the left column of Figure 5. Upon application of the AK, the tropospheric column $NO_2$ in the model simulation artificially increases in rural areas by +15.4%, while the urban $NO_2$ will artificially decrease. The latter due to most $NO_2$ being below 2 km due to large $NO_X$ emissions near the surface in urban areas where AK < 1.

Once the tropospheric averaging kernel is applied, it can be compared to the satellite directly (top row of Figure 5). In Dallas-Fort Worth and Houston, there are lower amounts of $NO_2$ in the model simulation in the most polluted areas of the city, but generally good agreement (+0.4%) when the five urban areas (Dallas, Fort Worth, Houston, San Antonio, Austin) are averaged together. In the rural areas of east Texas, there are slightly larger amounts (+10.7%) in the model simulation than as observed by TROPOMI, but these absolute differences are small. The largest disagreements between CAMx and TROPOMI occur in the vicinity of large point sources, which we discuss further in Section 3.4

While applying the averaging kernel to a regional model simulation is an appropriate way to compare model simulations with satellite data, it does so by artificially adjusting the high-resolution model simulation to be following the coarse resolution ($1.0° \times 1.0°$) of the TM5-MP model simulation used to originally process the AMF. Instead, incorporating the high-resolution model vertical profiles in the calculation of the AMF, while more computationally intensive, results in satellite measurements incorporating higher spatial resolution information; in urban areas this yields satellite measurements that have greater spatial heterogeneity.

In the middle row of Figure 5, we show a comparison between the model and the satellite with the CAMx-derived AMF. In this comparison, we get similar conclusions as mentioned earlier: the model has systematically smaller $NO_2$ amounts than TROPOMI in Dallas-Ft Worth and Houston, and larger amounts in rural areas. The agreement between the satellite measurement with a new AMF applied and model simulation is marginally different than when the averaging kernel is applied to the model simulation and compared to the satellite measurement directly. The percentage difference calculations differ primarily because the denominator (i.e., TROPOMI value) is a different magnitude in each case. We attribute this small difference to the rounding errors in the interpolation of the averaging kernel to the CAMx model pressure levels.

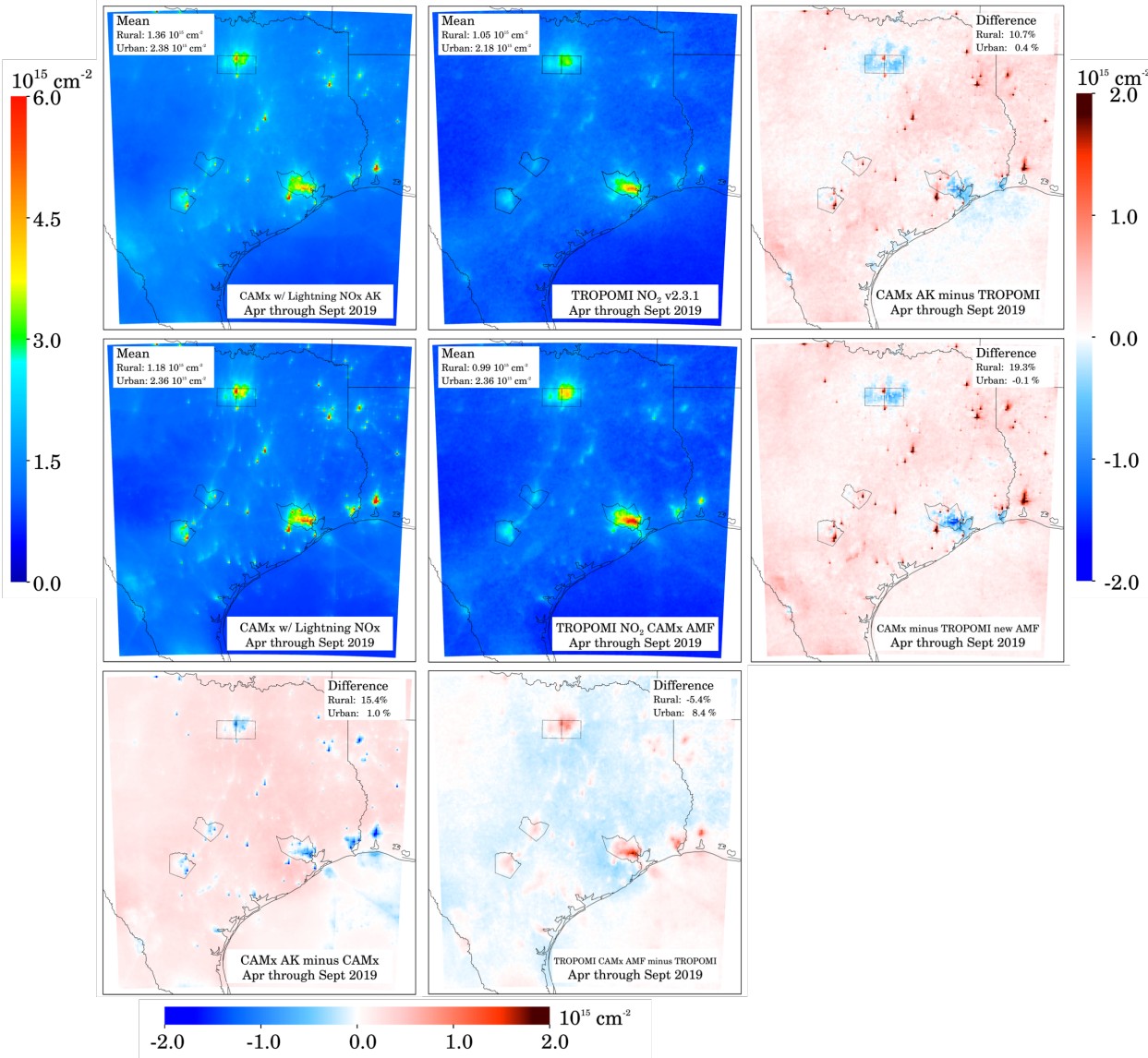

**Figure 5. (Top row)** NO₂ tropospheric vertical column amounts from CAMx and TROPOMI v2.3.1 re-processed with a priori profiles from the CAMx model with lightning NOx emissions, and difference averaged across April through September 2019. **(Middle row)** NO₂ tropospheric vertical column amounts from CAMx with the averaging kernel applied, the TROPOMI v2.3.1 product and difference averaged across April through September 2019. **(Bottom row)** Difference between top and middle rows. Areas with invalid TROPOMI data are similarly screened out from the model out on a daily basis. Urban area is defined as the five counties surrounding the largest five cities (Houston, Dallas, Fort Worth, San Antonio, and Austin). Rural area is everywhere outside those counties.

**3.4 Localized TROPOMI vs. CAMx NO₂ comparison**

We evaluate two versions of the TROPOMI seasonal average against the CAMx model simulation: TROPOMI v2.3.1 and TROPOMI v2.3.1 with CAMx AMFs. In Figure 6, a comparison of these satellite products versus CAMx are shown for four metropolitan areas: Dallas (DFW), San Antonio (SAT), Austin (AUS), and Houston (HOU). Comparing TROPOMI v2.3.1 to CAMx directly without application of the averaging kernel (which is not recommended) suggests a model high bias of +8.4% but moderately good association with each other ($r^2$=0.70). We then use the *a priori* profiles from the CAMx simulation to recalculate the AMF and find that the original model high bias in urban areas becomes a low bias of -0.1%, and becomes a larger low bias in the most polluted sections of the cities (consistent with our Discussion in Section 3.3). The low model bias is most pronounced in east Houston and the downtown area of Dallas. For Dallas-Fort Worth, there also appears to some spatial misallocation: NO₂ near the DFW airport is larger in the model than the satellite, while NO₂ in the downtown areas of Dallas and Fort Worth is smaller in the model than the satellite. In San Antonio and Austin, there is a small model overestimate, which becomes worse near the large point sources on the periphery of the city. Overall, however, there is generally good performance between CAMx NO₂ and TROPOMI NO₂, which is within 20% in most cases. The 20% is well within the expectation of TROPOMI accuracy and precision. The nonpoint $NO_X$ emissions input into the model simulation (e.g., mobile, nonroad, and area sources) generally are within the uncertainty of the satellite measurement, and we would not recommend a substantial alteration to the inventory for these sector emissions, except in the east Houston neighborhood. This exercise demonstrates the importance of utilizing the AMF when comparing satellite data to model simulations.

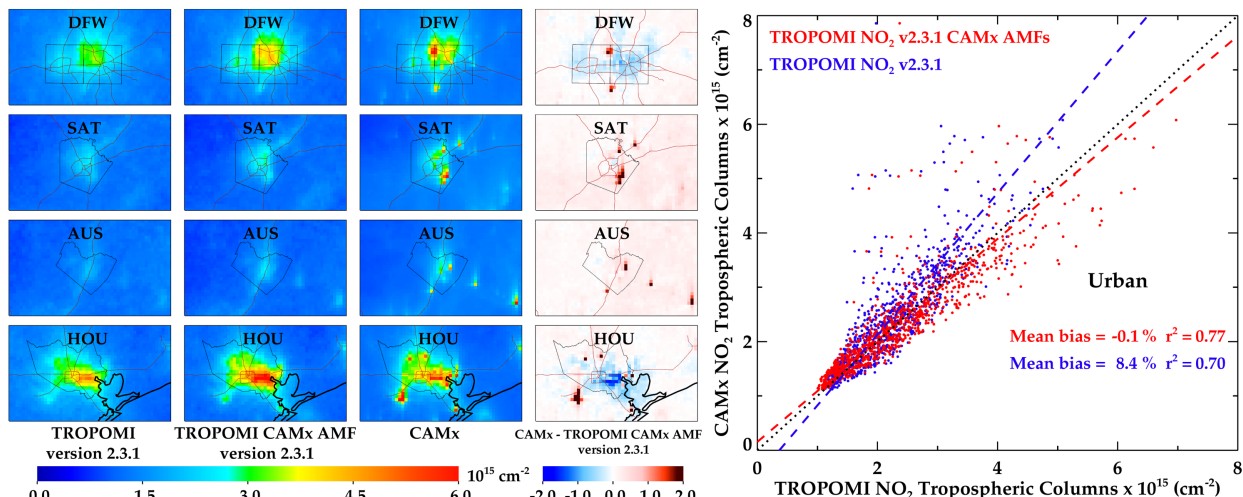

**Figure 6.** NO₂ tropospheric vertical column amounts averaged across April through September 2019 from TROPOMI, TROPOMI v2.3.1, TROPOMI v2.3.1 with new AMF, and CAMx for the largest four cities (Dallas, San Antonio, Austin and Houston). (Right) Scatterplot showing slope and correlation of various TROPOMI configurations and CAMx

To evaluate the performance of TROPOMI in observing point source emissions, we compare TROPOMI NO₂ measurements at the locations of three power plants with stack measurements: Martin Lake, Limestone and Sam Seymour (Figure 7). In each case, TROPOMI substantially underestimates NO₂ at the locations of these power plants

even when the new algorithm and recalculated AMF are both applied. We have previously found better agreement between TROPOMI $NO_2$ and the stack measurements for the Colstrip Power Plant in Montana and San Juan / Four Corner complex in New Mexico (Goldberg et al., 2019). The reason for the substantial disagreement in Texas is still unknown, but we do not believe this repudiates our prior evaluation for urban areas. $NO_X$ emissions from the power sector in the U.S. have declined by 76% between 2005 (3.63 million tons) and 2019 (0.86 million tons) (https://ampd.epa.gov/ampd/). At these lower emission rates, it appears that TROPOMI is having difficulty distinguishing $NO_2$ attributed to power plants from the background $NO_2$ concentrations especially in areas, such as Texas, with atmospheric conditions that cause short $NO_2$ lifetimes – rapid plume dilution, high oxidation capacity due to large amounts of VOCs and water vapor, and high solar elevation angles. Secondarily, the $NO_X/NO_2$ ratio in the model may be underestimated due to the 4 km grid cell size (Appendix C). The two power plants in New Mexico and Montana are located in areas with smaller background $NO_2$, lighter wind speeds, less VOCs and water vapor, and higher elevations; all of these factors cause the satellite sensor to be more sensitive to the $NO_X$ emissions. TROPOMI does not have the same difficulty over urban areas because the larger aggregated $NO_X$ emissions are more easily distinguishable from background concentrations. Please see Appendix D for a discussion on this topic. Future work should focus on evaluating the $NO_2$ from power plants and the $NO_X / NO_2$ ratio as the plume involves, such as *in situ* measurements from aircraft and ground-based vertical column instruments (e.g., Pandora (Herman et al., 2009)).

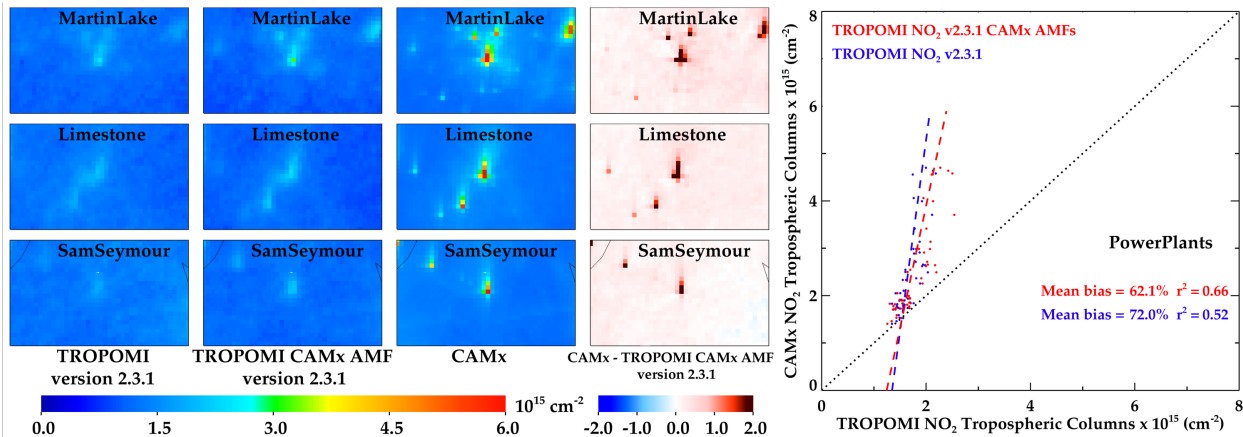

**Figure 7.** $NO_2$ tropospheric vertical column amounts averaged across April through September 2019 from TROPOMI v2.3.1, TROPOMI v2.3.1 with new AMF, and CAMx for the largest three power plants in East Texas (Martin Lake [Lat: 32.25 ° N, Lon: 94.58° W], Limestone [Lat: 31.42° N, Lon: 96.25° W], and Sam Seymour [Lat: 29.92° N, Lon: 96.75° W]). (Right) Scatterplot showing slope and correlation of various TROPOMI configurations and CAMx

## 4 Policy-relevant findings based on TROPOMI-model evaluation

### 4.1 TROPOMI $NO_X$ emissions

In order to calculate $NO_X$ emissions directly, we need to account for the $NO_2$ lifetime and $NO_2$ background concentrations. The first technique we use is the exponentially modified Gaussian (EMG) method. We first apply the EMG method to the CAMx simulations of the Limestone Power Plant (Latitude: 31.42° N, Longitude: 96.25° W) $NO_X$ plume. By comparing the known emissions with the inferred top-down emissions, we can evaluate assumptions in the

EMG model. The amount of $NO_X$ emissions input into the model within a 12 km radius of the facility are 9.8 Gg/yr. The top-down EMG method applied to the CAMx simulation yields a $NO_X$ emissions rate of 13.1 Gg/yr. The disagreement between the $NO_X$ emissions inventory (9.8 Gg/yr) and the inferred CAMx NOx emissions driven by the inventory (13.1 Gg/yr), must be due to incorrectly assumed effective wind speed likely driven by the meandering of the winds. Winds rarely have a consistent direction and instead meander due to boundary layer turbulence and frictional effects yielding a slower effective speed in the wind direction over long distances (>10 km). If we assume that the effective speed of the $NO_2$ plume to be 25% slower than the unidirectional wind speed, the inferred top-down emissions can be made to match the known emissions (9.8 Gg/yr).

Applying the CAMx-based effective plume speed to analysis of TROPOMI (25% slower than the unidirectional wind speed), we find that TROPOMI $NO_2$ v2.3.1 product yields an estimated $NO_X$ emissions rate of 5.2 Gg/yr, and increased to 6.0 Gg/yr when using the TROPOMI v2.3.1 algorithm with a recalculated AMF (Table 2 & Figure 8). Even with all known corrections applied, it appears that TROPOMI is not capturing the full magnitude of $NO_X$ emissions from the power plant and vicinity (9.8 Gg/yr) which is consistent with the discussion in Section 3.4.

**Table 2.** $NO_X$ emission rates for Dallas – Fort Worth and the Limestone Power Plant from the TCEQ Emissions Inventory and various iterations of the TROPOMI $NO_2$ algorithm

| Data Source | Data Source Type | Dallas-Fort Worth NOx emissions (Gg/yr) | Limestone PP NOx emissions (Gg/yr) |
|---|---|---|---|
| TCEQ Projected 2020 Inventory | Bottom-up | 55 | 9.8 |
| TROPOMI $NO_2$ v2.3.1 | Top-down | $56 \pm 20$ | $5.2 \pm 1.9$ |
| TROPOMI $NO_2$ v2.3.1 CAMx AMFs | Top-down | $62 \pm 22$ | $6.0 \pm 2.2$ |

For the Dallas – Fort Worth area, if we apply the same method to the CAMx simulation, we get an effective $NO_X$ emissions rate of 55 Gg/yr from the metropolitan area. This is equivalent to the $NO_X$ emissions aggregated within a 47 km radius of the Dallas – Fort Worth metropolitan area (Latitude: 32.85º N, Longitude: 96.95º W), and is roughly equivalent to two-sigma of the Gaussian plume ($\sigma$ = 23.7 km).

Using the TROPOMI v2.3.1 algorithm, we calculate a top-down $NO_X$ emissions rate of 56 Gg/yr and increased to 62 Gg/yr when a CAMx AMF is used (Table 2 & Figure 8). The difference between the 62 Gg/yr calculated directly from the TROPOMI v2.3.1 with a recalculated AMF and the 55 Gg/yr effective emissions rate from CAMx represents a small 13% low bias that is within the uncertainty of the satellite and the assumptions made to facilitate the comparison. The technique was applied to other urban areas, but those cities have large point sources at the periphery of the urban areas which adversely affected the calculation of the effective $NO_2$ lifetime needed to calculate the $NO_X$ emissions.

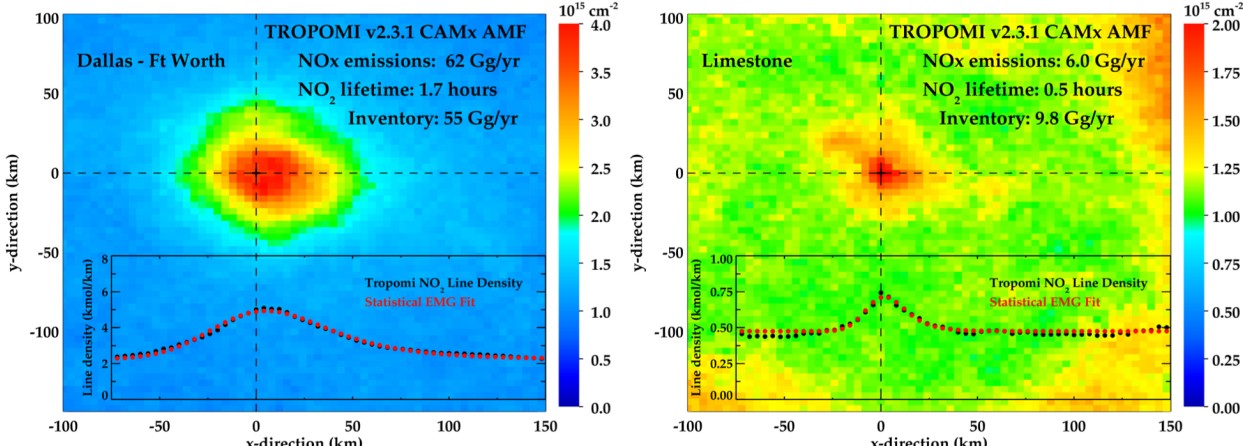

**Figure 8.** EMG method to derive NO$_X$ emissions from the TROPOMI NO$_2$ v2.3.1 with CAMx AMFs applied to (left) Dallas-Fort Worth and (right) Limestone Power Plant. The colorbar for the right panel is halved to better show the NO$_2$ plume near Limestone. ERA5 100-m winds are used to rotate daily TROPOMI NO$_2$ plumes.

The top-down approach can also calculate effective NO$_2$ lifetimes. Most top-down methods fit both the effective NO$_2$ lifetime and NO$_X$ emissions simultaneously and therefore have a "seesaw relationship" – as lifetime increases, NO$_X$ emissions decrease given a constant NO$_2$ burden. Here, we visually inspect the plume to ensure that the NO$_2$ effective lifetime is reasonable given the plume decay before proceeding. For Dallas – Fort Worth, the method calculates an effective NO$_2$ lifetime of 1.7 hours. The same approach applied to CAMx yields an effective NO$_2$ lifetime of 1.1 hours. This suggests that the effective NO$_2$ lifetime in CAMx is too short. The effective lifetime is a function of the chemical lifetime and dispersion lifetime (de Foy et al., 2014):

$$\frac{1}{\tau_{effective}} = \frac{1}{\tau_{chemical}} + \frac{1}{\tau_{dispersion}} \tag{4}$$

The effective lifetime could be increased in a model simulation by increasing the NO$_2$ chemical lifetime (e.g. slower photolysis, slowing the NO$_2$+OH reaction rate, faster recycling of NO$_Z$ (NO$_Z$ = Alkyl nitrates, PAN, and HNO$_3$) back to NO$_2$) or by increasing vertical mixing (less plume meandering at higher altitudes due to less surface frictional effects). Chemical NO$_2$ lifetimes are well-constrained by laboratory studies, so we hypothesize that too slow vertical transport may be the primary culprit for this disagreement, and is also suggested by the analysis presented in Figure 3, which suggests a model low bias in the free troposphere using measurements from the SEAC4RS campaign. Future vertical NO$_2$ measurements separated by altitude will be critical to answering this question.

The total error associated with the magnitude of the top-down versus bottom-up comparison is calculated to be 36%, and is the sum of the quadrature of five potential sources of error: the tropospheric vertical column measurement in urban areas (20%), the wind speed & direction (25%) (Appendix B), the "clear-sky" bias (10%) which for these purposes is a result of emissions being different on clear-sky days compared to cloudy days, the NO$_X$/NO$_2$ ratio (10%) (Appendix C), and the random error of the statistical EMG fit (10%) (de Foy et al., 2014). This total uncertainty is approximately 20% smaller than similar methods using OMI. For further information on this method

or the uncertainties associated with this method, please see other literature (de Foy et al., 2014; Goldberg et al., 2019a; Lu et al., 2015; Verstraeten et al., 2018).

We then test the flux divergence method (Beirle et al., 2019, 2021; de Foy and Schauer, 2022) on the same two sources: Dallas and Limestone Power Plant. We apply the flux divergence method to the native TROPOMI pixels rather than a re-gridded version of the data. Figure 9 shows that TROPOMI columns distinguish between a large hotspot over Dallas and a smaller one over Fort-Worth. For the Dallas urban area, the algorithm identified 11 separate source regions which were each represented by a separate two-dimensional Gaussian. The flux divergence method was able to resolve source regions with better detail, with estimates for some of the individual point sources and sub-areas within Dallas. In particular the area including the Dallas-Fort-Worth International Airport appears as a distinct source area. In Table 3, we show the $NO_X$ emissions aggregated for these two sources, using both an infinite $NO_2$ lifetime and the effective "short" $NO_2$ lifetime provided by the EMG method ($\tau$ =1.7 h for Dallas-Fort Worth and $\tau$ = 0.5 h for Limestone PP). The results from the flux divergence method are consistent with the results from the EMG method (i.e., Dallas $NO_X$ is within 20% and power plants $NO_X$ are biased low by ~65%) provided that a short $NO_2$ lifetime is assumed.

**Table 3.** $NO_X$ emission rates for Dallas – Fort Worth and the Limestone Power Plant from the TCEQ Emissions Inventory and various iterations of the Flux Divergence Method using the TROPOMI $NO_2$ v2.3.1 algorithm

| Data Source | Dallas-Fort Worth NOx emissions (Gg/yr) | Limestone PP NOx emissions (Gg/yr) |
|---|---|---|
| TCEQ Projected 2020 Inventory | 55 | 9.8 |
| TROPOMI $NO_2$ v2.3.1, Infinite $NO_2$ Lifetime | 24 ± 9 | 1.6 ± 0.4 |
| TROPOMI $NO_2$ v2.3.1, Short $NO_2$ Lifetime | 62 ± 16 | 3.4 ± 1.1 |

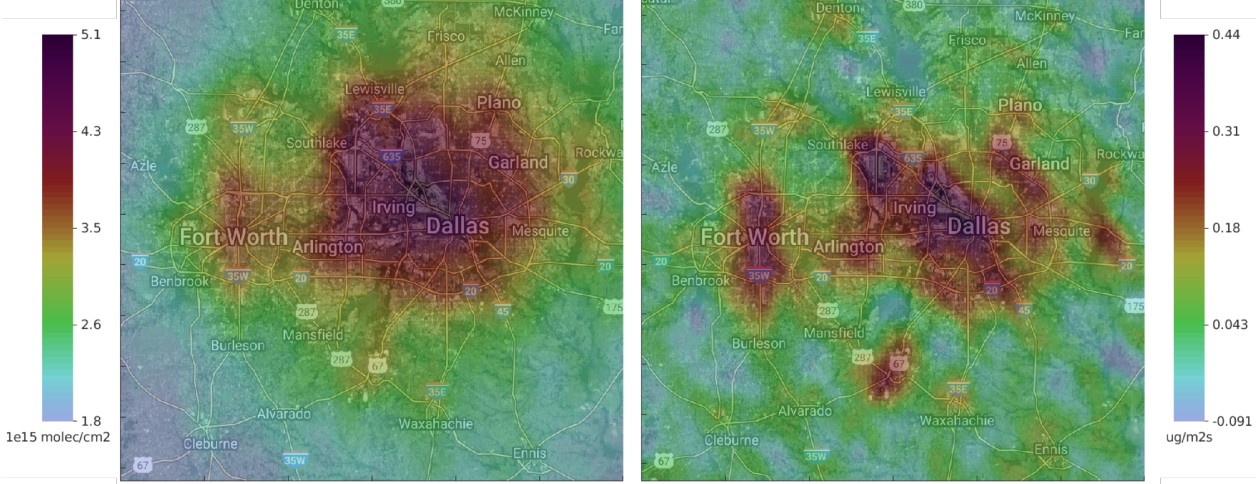

**Figure 9.** Oversampled TROPOMI $NO_2$ in the Dallas-Fort Worth metropolitan areas using the (left) tropospheric vertical columns and (right) the flux divergence of the tropospheric vertical columns. Image underlaid is from © Google Earth.

**4.2 Evaluating ozone sensitivity using the HCHO-NO₂ ratio**

Satellite observations of formaldehyde (HCHO) can be combined with $NO_2$ to determine the ozone sensitivity to $NO_X$ emissions using the formaldehyde to nitrogen dioxide column density ratio (FNR) (Duncan et al., 2010; Jin et al., 2017; Jin and Holloway, 2015; Martin et al., 2004). HCHO may be used to estimate short-lived VOC emissions, anthropogenic and biogenic combined, which often quickly oxidize to HCHO in the presence of sunlight and the hydroxyl (OH) radical (Wolfe et al., 2016; Zhu et al., 2017). In a similar manner to $NO_2$, column HCHO can be compared to chemical transport models in order to better understand the spatial variability of VOC emissions. Harkey et al. (2020) found that a regional model captured the general spatial and temporal behavior of satellite estimates, but tended to underestimate column HCHO in the western U.S. TROPOMI HCHO measurements have been rigorously evaluated using ground-based spectrometers and the v1.1 algorithm was found to be biased low by approximately 25% (de Smedt et al., 2021).

We first compare column HCHO comparison between CAMx and TROPOMI. Tropospheric column TROPOMI HCHO measurements using the v1.1 algorithm are biased low by approximately 25% (De Smedt et al., 2021). We then create a bias-corrected (b-c) product (multiply by a factor of 1.25) to account for this low bias. In Figure 10, we compare the operational TROPOMI HCHO v1.1 product and TROPOMI HCHO v1.1 b-c product to CAMx tropospheric columns amounts with and without the averaging kernel sampled at coincident timeframes. Since HCHO spatial patterns have less heterogeneity than $NO_2$, due to a large fraction of HCHO originating from biogenic precursors during warm season months, column HCHO amounts are less sensitive to the application of the AK than with $NO_2$. The difference between CAMx and CAMx with the averaging kernel applied is ±2.5% for areawide averages. CAMx underestimates HCHO in Central and Western Texas, but in Eastern Texas the magnitude and spatial patterns match better. The model bias is -7.9% in Eastern Texas and -25.0% in Central Texas compared to the TROPOMI HCHO v1.1 b-c product. This model bias, in both cases, is within the uncertainty of the satellite retrieval.

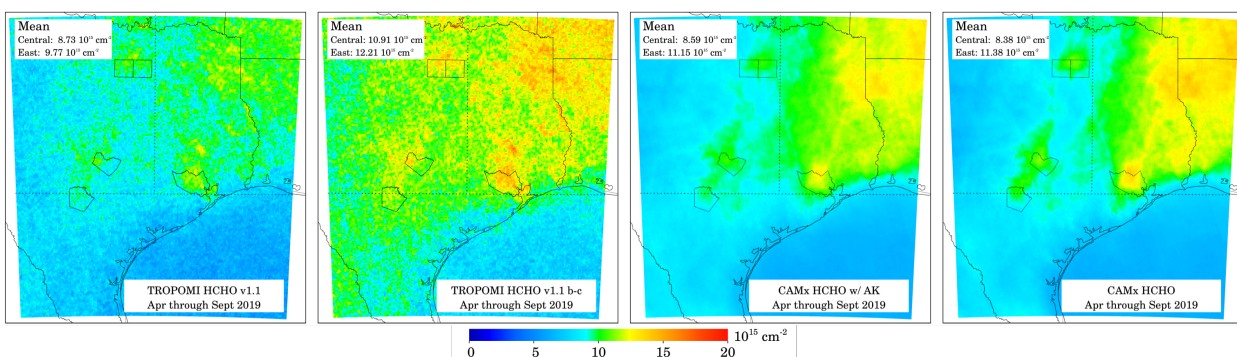

**Figure 10.** HCHO tropospheric vertical column amounts averaged across April through September 2019 from (a) TROPOMI, (b) TROPOMI bias-corrected, (c) the CAMx regional model with TROPOMI averaging kernel (AK) applied, and (d) CAMx without the AK applied. All model information is shown at the coincident TROPOMI overpass time (~19 UTC). Areas with invalid TROPOMI data are similarly screened out from the model out on a daily basis. The Eastern and Central Texas areas are denoted by the dashed lines

We apply the FNR to TROPOMI and CAMx to determine how well CAMx is representing ozone formation regimes. Initial studies showed that when the FNR in a region exceeds 2, the ozone formed is considered to be limited by the amount of $NO_x$ present in the air. When the FNR is below 0.5, the ozone formed is considered to be limited by the amount of VOCs. Ratio values between 0.5 and 2 indicate sensitivity to both $NO_x$ and VOCs (Duncan et al., 2010).

More recent studies have demonstrated that the upper bound of the transitional regime could be as high as 4 (or even higher) depending on regional characteristics (Jin et al., 2017, 2020; Schroeder et al., 2017).

For this analysis, using the v1.1 HCHO and v1.3 $NO_2$ algorithms is sufficient, since both products have similar biases related to the cloud schemes that may cancel out when a ratio is calculated. We use a value of 4 to indicate the transition between $NO_X$ and VOC sensitivity, while simultaneously noting that this value should not be static in all scenarios.

In Figure 11, the ratios from the satellite and model for each area are shown directly on the plot.

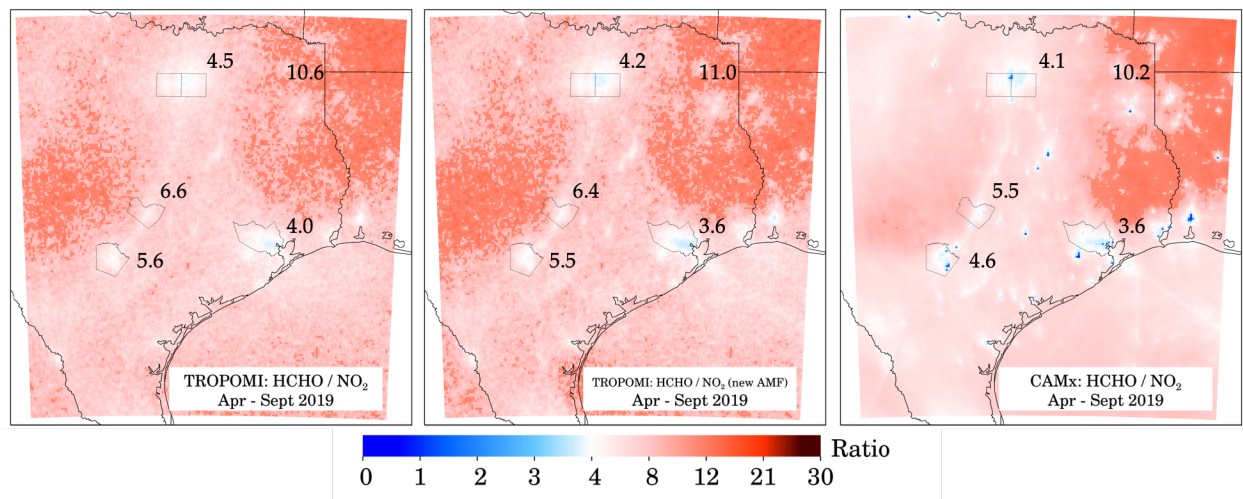

**Figure 11.** Formaldehyde – $NO_2$ – Ratio (FNR) in Texas averaged across April through September 2019 using the (left) operational TROPOMI products (center left) operational TROPOMI HCHO product and TROPOMI $NO_2$ product with new AMFs and (right) CAMx column amounts. Only CAMx data coincident with the overpass time and

valid TROPOMI pixels are included. The ratios from the satellite and model for each area are shown directly on the plot.

On a regional scale, there is excellent spatial agreement between the satellite and model. Ozone formation conditions are $NO_X$-limited (FNR>4) throughout the vast majority of Texas; other studies have found similar conclusions within the last five years (Jin et al., 2020; Koplitz et al., 2021). Only along the Houston ship channel, near the DFW airport,

and in the presence of undiluted power plant plumes are conditions potentially in the transitional regime. When aggerated on an urban scale, the model ratio values are marginally lower than the satellite derived ratios, especially in San Antonio and Austin. This model low bias is improved when the AMF of the $NO_2$ product is recalculated. Consistent with the analyses presented in Sections 3.3 and 3.4, the model appears to be capturing both the HCHO and $NO_2$ spatial patterns with satisfactory performance and therefore the ozone production regimes are also captured well.

The only areas of strong disagreement are in the presence of power plant plumes and large point sources, which TROPOMI appears to be not fully characterizing.

The downside of low-earth orbiting instruments is the consistent measurement during the early afternoon. This early afternoon measurement time coincides with; 1.) a temporary dip in $NO_X$ emission rates, which are largest in the early morning and late afternoon, 2.) the peak of the biogenic VOC emissions, which often peak at the time of the maximum daily 2-m temperature, and 3.) stronger photolysis rates which affect both $NO_2$ and HCHO.

We use the CAMx model to investigate the temporal variation in the FNR. In Figure 12, we show diurnal cycles of column $NO_2$, column HCHO, and the FNR. The $NO_2$ diurnal cycle has a minimum in the early afternoon driven mostly by the higher photolysis rates and secondarily by the relatively lower $NO_X$ emission rates compared to the early morning and late afternoon. HCHO has broad peak in the afternoon, which is likely related to biogenic emissions and secondary formation. However, the HCHO diurnal cycle is flatter than we expected; this may be due to model difficulties in representing complex VOC chemistry for secondary HCHO production (Schroeder et al., 2016; Schwantes et al., 2022).

According to CAMx, the FNR has a temporary maximum in urban areas around 14:00 local time and a minimum around 8:00 local time, with a secondary minimum around 20:00 local time. In the rural areas of East Texas, the variation of the FNR is even more substantial than in the urban areas, and even in these rural areas, ozone production might be VOC-limited during early morning hours. Therefore, an early afternoon satellite measurement suggesting $NO_X$-limited conditions does not eliminate the possibility of VOC-limited ozone formation conditions in the early morning. This suggests that targeted VOC controls in urban areas of Texas between 6:00 – 10:00 local time could be an effective way to further reduce ozone concentrations, in addition to expanded $NO_X$ controls at all hours. Upcoming observations from the Tropospheric Emissions Monitoring of POllution (TEMPO) instrument, which will be located in geostationary orbit, which further help answer this question.

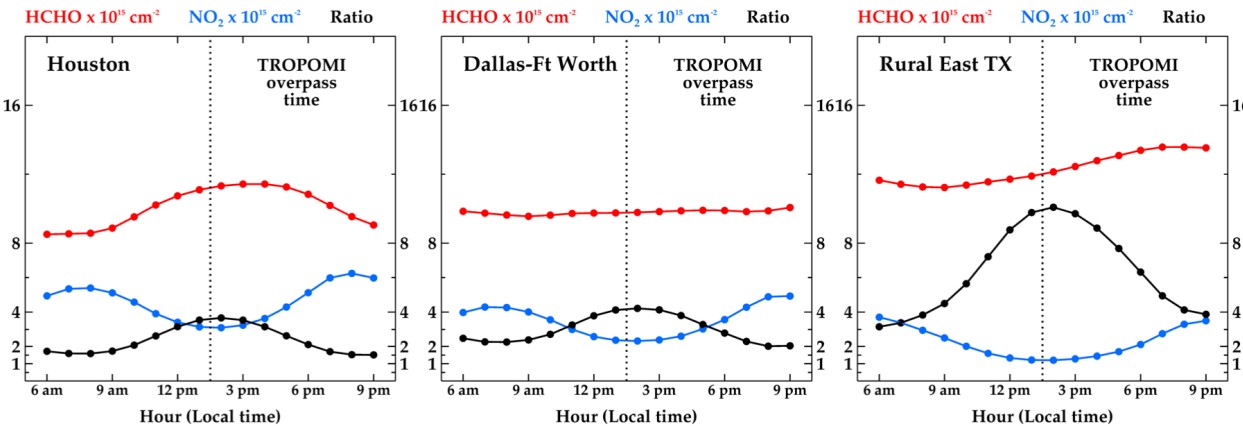

**Figure 12.** Diurnal cycles of column $NO_2$, column HCHO, and the HCHO/$NO_2$ ratio from CAMx for these regions in our model domain: Houston, Dallas-Fort Worth and Rural East Texas (Cass County). The approximate TROPOMI overpass time of 13:30 local time is denoted by the dotted line.

**5 Conclusions**

In this study, we find that TROPOMI $NO_2$ columns offer a valuable means to validate NOx emissions inventories, with important limitations. When using locally resolved inputs, simulated urban $NO_2$ columns in Texas agree with TROPOMI to within 20% in most areas. Using data from the newest TROPOMI $NO_2$ algorithm (v2.3.1) generally

showed better agreement with the model. We find some evidence that $NO_X$ emissions in certain sections of Dallas – Fort Worth, TX and Houston, TX may be underestimated, but the underestimates are within the uncertainty of the methods presented herein.

In the rural areas of east Texas, we find generally good agreement to within 20% in most circumstances between the model and TROPOMI $NO_2$, when lightning $NO_X$ emissions are included. In rural regions of east Texas, >50% of the

column $NO_2$ appears to be above 2 km in altitude demonstrating the influence of the free tropospheric $NO_2$, including lightning. Lightning $NO_X$ emission can represent up to 24% of the column $NO_2$ in our east Texas domain, and presumably would be larger in more isolated tropical regions. Since free tropospheric $NO_2$ has an outsized effect in rural areas, it is critical to have an accurate estimate of free tropospheric $NO_2$ before conducting a model to satellite comparison in these regions (Shah et al., 2022). More aircraft measurements between the top of the boundary layer

and the stratosphere-troposphere interface would be helpful to better understand and quantify free tropospheric $NO_2$.

Over large power plant plumes, however, we find statistically significant differences between the model and satellite measurements. Because the $NO_X$ emissions from these power plants are directly measured, we conclude that TROPOMI cannot distinguish $NO_2$ attributed to power plants from the background $NO_2$ concentrations in Texas. This limitation may be due to short $NO_2$ lifetimes characteristic of that region, and secondarily the $NO_X/NO_2$ ratio in the 4

km model simulation. More work should be dedicated to investigating $NO_2$ and $NO_y$ partitioning near power plant plumes, including aircraft and vertical profilers (e.g. Pandora).

In our comparison between TROPOMI and modeled HCHO, we find excellent agreement in far eastern Texas and the Ozarks, but an underestimate in central Texas. This is consistent with Harkey et al. (2020), which showed a model underestimate in the Western U.S. More work should be done to evaluate HCHO and VOCs in areas with assumed

small amounts of biogenic emissions.

In a last step, we evaluate the ozone formation regimes at the time of the early afternoon TROPOMI overpass. We find that ozone production is NOx-limited almost everywhere in the domain except near the Houston Ship Channel, near the DFW airport, and in the presence of power plant plumes. There are likely NOx-saturated ozone formation conditions in the early morning hours that TROPOMI cannot observe.

We are encouraged by the future observational strategies that could help tackle some of the remaining questions presented herein. In early 2023, TEMPO will be acquiring column $NO_2$ and HCHO measurements during all daylight hours in the presence of low amounts of clouds. When coupled with the current ground monitoring network, this will

elucidate some of the unknown $NO_2$ and HCHO diurnal cycles, giving us more confidence in our understanding of $NO_X$ emissions, $NO_2$ chemistry, and satellite retrievals.

**Appendix A. CAMx model simulation performance**

We evaluated CAMx NOx and ozone surface concentrations using data collected at TCEQ Continuous Air Monitoring Stations (CAMS). We evaluated performance by five geographical sub regions: Austin, San Antonio, Waco, Tyler, and Dallas-Fort Worth. $NO_x$ monitors deployed for routine monitoring have limitations for $NO_2$. These monitors measure NO and consequently $NO_2$ is chemically converted to NO for measurement. The converter also captures other compounds including peroxyacyl nitrate (PAN) and a portion of $HNO_3$ (Dickerson et al., 2019). These NOx monitors have a detection limit of around 1 ppb but differentiation between NO and $NO_2$ is less accurate near the detection limit. Therefore, we compare both CAMx NOx (i.e., NO + $NO_2$) and NOy (i.e., NO + $NO_2$ + PAN compounds + $HNO_3$) to monitored NOx in Figure A1. Hourly ozone measurements were aggregated to 8-hour maximum daily averages (MDA8) and hourly $NO_2$ measurements were aggregated to early afternoon averages (12-3 PM CST) to correspond with TROPOMI overpass time.

Figure A1 displays the $O_3$ and $NO_2$ performance in the CAMx simulation compared to ground monitors. High observed NOx detected by ground monitors in urban areas (e.g. > 10 ppb) are not resolved at the 4 km CAMx horizontal grid resolution. As discussed in (Souri et al., 2022), care is needed when comparing pointwise measurements to concentrations spatially averaged over large (>1 km) grid cells. For example, Dallas Hinton St (CAMS 0401) is located 0.9 km from a major freeway interchange and 200 m from a busy road (Mockingbird Lane). In contrast, Tyler Airport (CAMS 0082) is in a rural location removed from busy roads and the nearby airport is regional and not highly trafficked. When compared with monitored NOx in less polluted areas (i.e. < 10 ppb), CAMx NOx tends to be lower than measured NOx whereas CAMx NOy tends to be higher than measured NOx. We therefore conclude that CAMx is consistent with the ambient NOx measurements within limitations of the monitoring equipment capabilities and siting.

We present similar scatter plots for maximum daily 8-hour average (MDA8) ozone in Figure A1. CAMx shows skill in identifying low and high ozone days, with $R^2$ values from 0.56 (Austin) to 0.61 (Tyler). CAMx displays a positive ozone bias across all five regions, with mean bias (MB) ranging from 4.8 ppb (Waco) to 10.1 ppb (San Antonio). Emery et al. (2017) defines the criteria standards for MDA8 ozone as ± 15% for normalized mean bias (NMB) and < 25% for normalized mean error (NME). Only Waco and Dallas-Fort Worth meet the criteria standard for NMB, while all regions except San Antonio meet the criteria standard for NME.

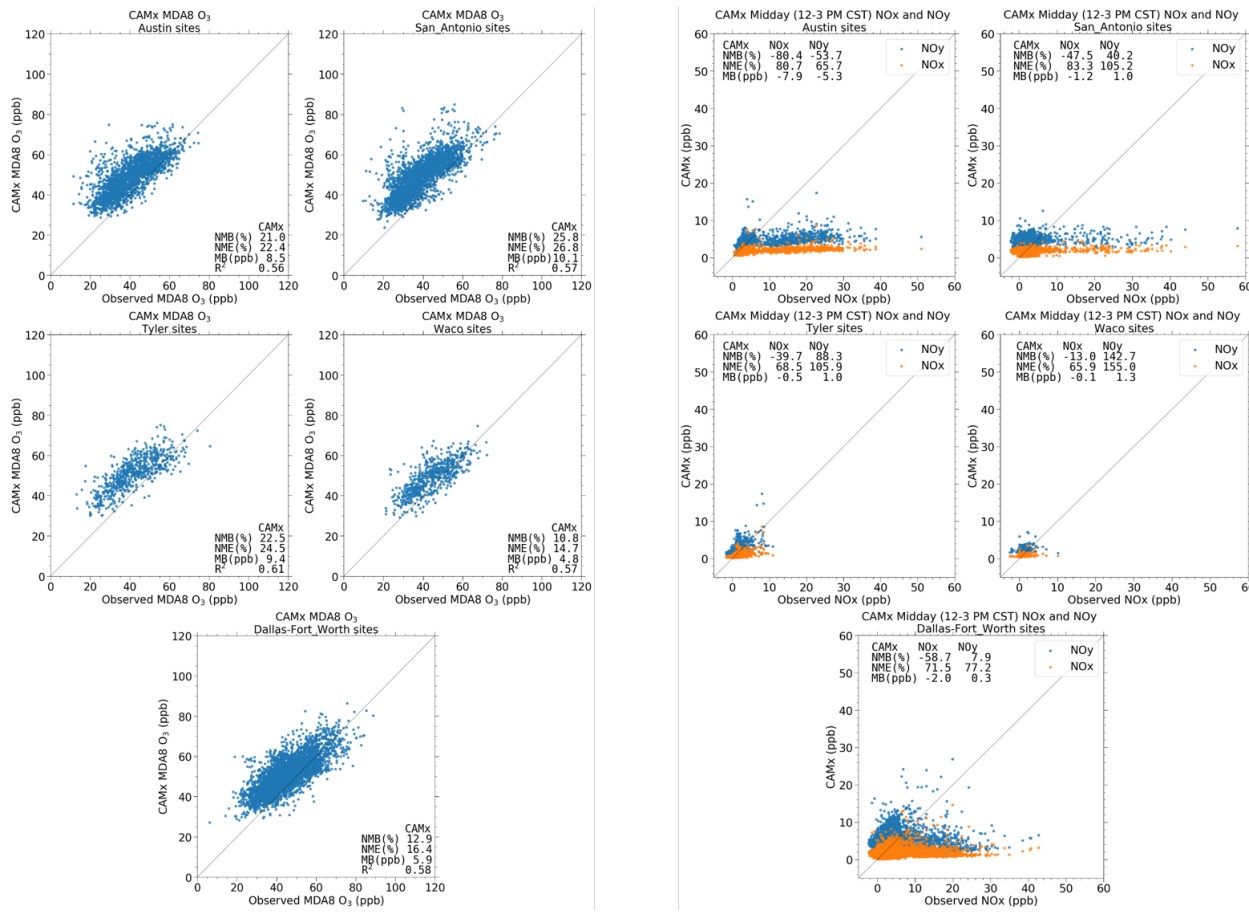

**Figure A1.** CAMx model performance for (left) maximum daily averaged 8-hour ozone (MDA8 O₃) and (right) midday 12 – 3 PM local time NO$_X$ and NO$_y$. Model output is compared to the EPA AQS ground observations for five regions of interest in our east Texas domain (Austin, San Antonio, Tyler, Waco, and Dallas-Fort Worth)

**Appendix B. ERA5 Winds**

To justify, the use of the ERA5 100-m winds (as opposed to another vertical level or interval), we use the $NO_2$ column information from CAMx to determine the weighted column mid-point. Using the shape profiles described in Figure 3, we find that 50% of the tropospheric $NO_2$ column in the Dallas - Fort Worth area is below 227 m in altitude (and therefore 50% is above this); this is the weighted column mid-point. Using the WRF simulation, we find that the 100-m wind speed is 6% slower than the 227 m wind speed in Dallas – Fort Worth. However, as we discuss in Section 4.1, errors due to wind meandering (~25%) are far more critical.

We can then apply uncertainty bounds to this. In the most polluted sections of the city, the column mid-point would be lower (10s of m), and in the least polluted sections of the city the column mid-point can be as high as 500 m. While neither of these are appropriate for an areawide average, they can constrain the uncertainties of the column midpoint. Using the WRF simulation, we find that winds at 500 m are 15% larger and surface winds are 24% lower than the 100 m wind speed.

**Table B1. Wind speeds at Dallas – Fort Worth for the April – Sept 2019 average at various vertical levels in comparison to the 100-m wind speed**

| Wind fields | Ratio to 100-m wind speed |
|---|---|
| 10-m winds | 0.76 |
| 100-m winds | 1 |
| 227-m winds | 1.06 |
| 500-m winds | 1.15 |

**Appendix C. NOₓ/NO₂ ratio**

To further investigate whether the $NO_X/NO_2$ ratio used in our study is appropriate, we probe the CAMx simulation to calculate the $NO_X/NO_2$ ratio for the partial columns below 2 km. The $NO_X/NO_2$ ratio above 2 km is inappropriate for use in the EMG method since the column above 2 km represents "background conditions" and is subtracted out when using the EMG method.

The $NO_X/NO_2$ ratio for the partial column below 2 km in urban areas is $1.31 \pm 0.02$ (Dallas: 1.33, Austin 1.30, San Antonio: 1.32, Houston: 1.295). For urban areas, this represents an uncertainty in the $NO_X/NO_2$ ratio of less than 10%. Our original assumption of using a $NO_X/NO_2$ ratio of 1.32 is warranted.

However, the $NO_X/NO_2$ ratio can vary more substantially near large point sources. In the grid cells of the large point source itself, the $NO_X/NO_2$ ratio can be as large as 1.52. It is possible that the $NO_x/NO_2$ ratio in the model may be underestimated due to the emissions being equally spread out across the 4 km grid cell. $NO_x/NO_2$ ratios can be as large as 2 within 100 m downwind of major $NO_X$ sources, especially under low ozone (< 30 ppb) conditions (Kimbrough et al., 2017). However, further downwind (>4 km) of these large point sources, the $NO_X/NO_2$ ratio quickly converges back to a value of ~1.31.

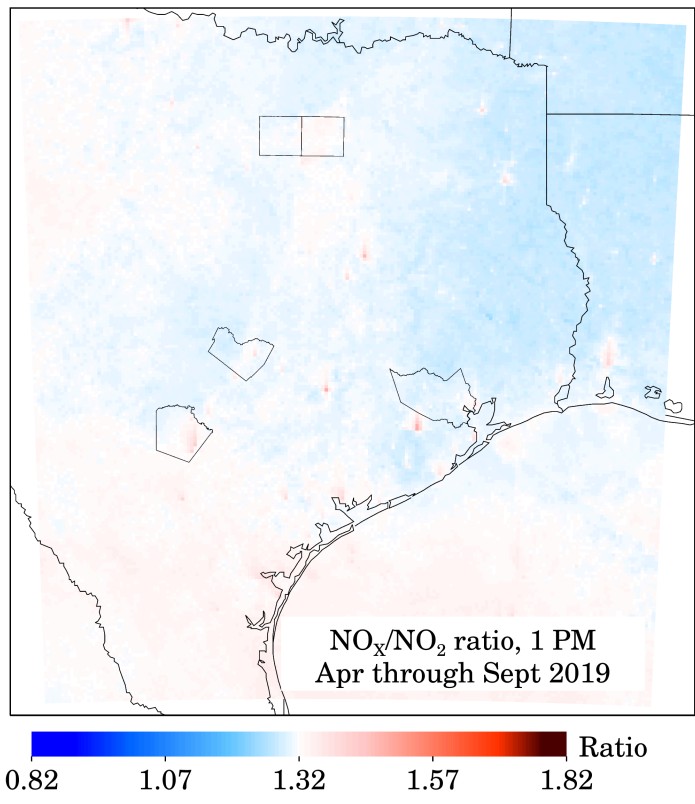

**Figure C1.** The $NO_X/NO_2$ ratio at 1 PM local time for April − Sept 2019 for partial $NO_2$ columns below 2 km in altitude, as simulated by CAMx.

**Appendix D. Daily TROPOMI NO₂**

Daily images of the TROPOMI $NO_2$ vertical column densities are shown in Figure D1. The top set of panels show the daily images over Dallas – Fort Worth during July 2019. These daily images document that a $NO_2$ plume can be observed on every day in which there are no clouds. We also plot the daily ERA5 wind speed and direction on each daily panel. ERA5 winds appear to correctly identify the urban plume direction on each day.

The middle and bottom set of panels (Martin Lake, TX and Colstrip, MT respectively), demonstrate the capability of TROPOMI in observing daily plumes from power plants during July 2019. For Colstrip (13,600 tons NOx/yr), a plume signature can be visually located on every cloud-free day. However, in Martin Lake, TX (9,500 tons NOx/yr), a plume signature cannot be visually located on every cloud-free day, even though $NO_X$ emissions are of a similar order of magnitude as Colstrip. This suggests that the location and atmospheric conditions in Texas are causing TROPOMI to not fully observe Martin Lake's $NO_X$ emissions.

**Dallas – Fort Worth, TX**

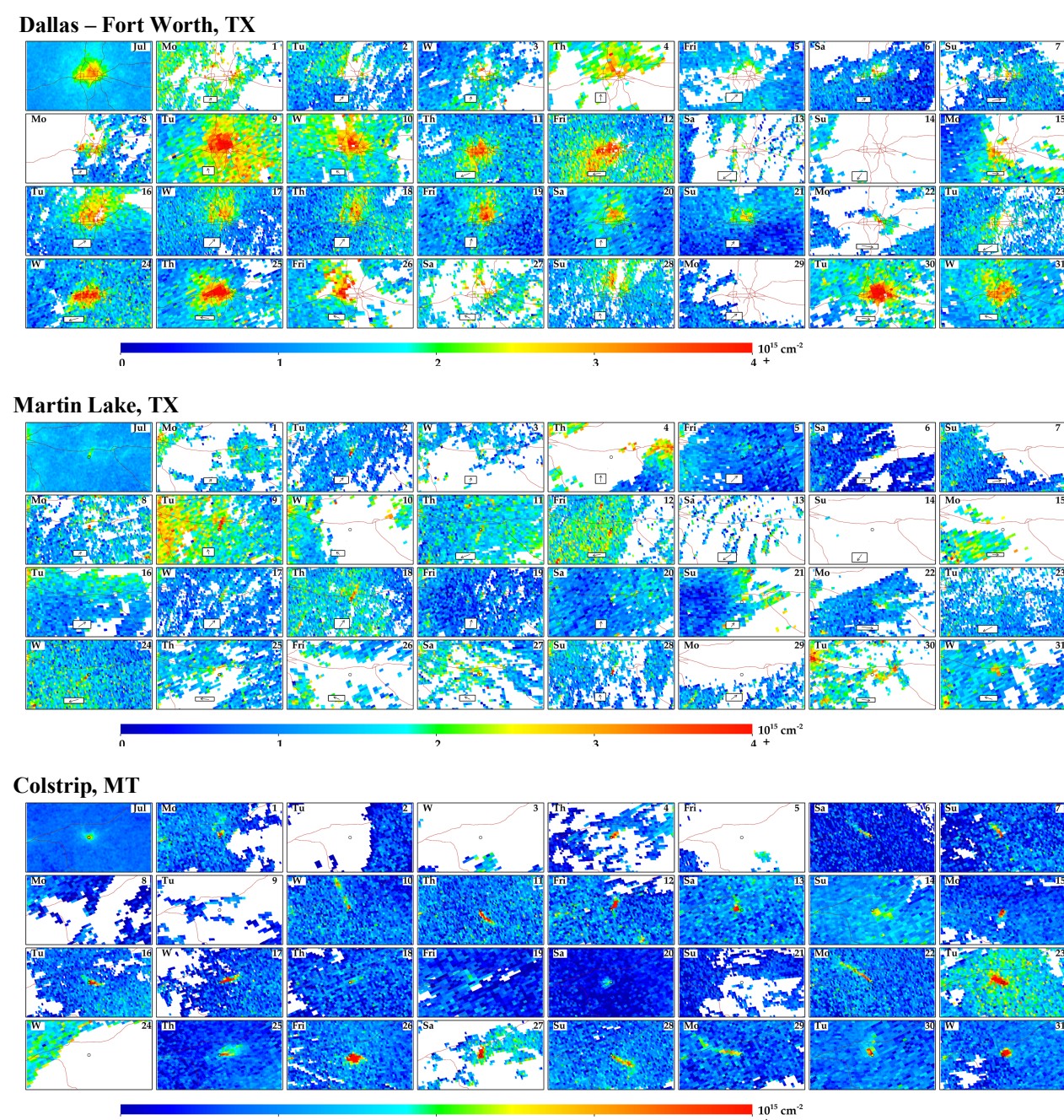

**Martin Lake, TX**

**Colstrip, MT**

**Figure D1.** Daily TROPOMI $NO_2$ vertical column densities over three locations (Dallas – Fort Worth, Martin Lake Power Plant, and Colstrip Power Plant) during each day of July 2019; the July 2019 monthly average is denoted in the top left panel of each location aggregate.

**Data availability**

TROPOMI NO$_2$ v1.3 data (doi: 10.5270/S5P-s4ljg54) and TROPOMI HCHO v1.1 data (doi: 10.5270/S5P-tjlxfd2) can be freely downloaded from the Copernicus Open Access Hub (https://s5phub.copernicus.eu/dhus/) or NASA Earthdata Hub (https://disc.gsfc.nasa.gov/datacollection/S5P_L2__NO2____1.html &

https://disc.gsfc.nasa.gov/datacollection/S5P_L2__NO2____HiR_1.html;
https://disc.gsfc.nasa.gov/datacollection/S5P_L2__HCHO___1.html  &
https://disc.gsfc.nasa.gov/datacollection/S5P_L2__HCHO___HiR_1.html). TROPOMI NO$_2$ v2.3.1 data can be freely downloaded from the S5P-PAL Data Portal (https://data-portal.s5p-pal.com/products/no2.html). NASA SEAC4RS data can be downloaded from NASA data archive (doi: 10.5067/Aircraft/SEAC4RS/Aerosol-TraceGas-

Cloud), and was acquired by the UC-Berkeley Cohen research team. ERA5 re-analysis hourly data on single levels (doi: 10.24381/cds.adbb2d47) can be downloaded from Copernicus Climate Data Store (https://cds.climate.copernicus.eu/#!/home). IDL code to re-grid and process the data is available upon request.

**Author contributions**

DG processed the satellite data, produced most of the figures, and wrote the manuscript. MH, BdF, and LJ provided
guidance processing the satellite data and helped to develop several of the figures. JJ performed the model simulation, processed the model output, and compared the model simulation to the ground monitors. GY and TH conceptualized the study, obtained funding, and provided guidance and feedback throughout. All authors helped to edit the text of the manuscript.

**Competing interests**

The contact author has declared that neither they nor their co-authors have any competing interests.

**Acknowledgments**

This preparation of this manuscript was funded by a grant (20-020) from the Texas Air Quality Research Program (AQRP) at the University of Texas at Austin through the Texas Emission Reduction Program (TERP) and the Texas Commission on Environmental Quality (TCEQ). The findings, opinions, and conclusions are the work of the authors
and do not necessarily represent findings, opinions, or conclusions of the AQRP or the TCEQ. The authors would like to thank the reviewers at TCEQ for their input on the manuscript. We appreciate feedback from Dr. Ron Cohen on the usage of the SEAC4RS data and Dr. Ted Russell on the intercomparison near power plants. This project was also funded by grants from the NASA Health and Air Quality Applied Sciences Team (HAQAST) (80NSSC21K0511), NASA Health and Air Quality (HAQ) (80NSSC19K0193) and the NASA Atmospheric
Composition Modeling and Analysis Program (ACMAP) (80NSSC19K0946). For the AQRP funding of this project, Dr. Goldberg was paid as a consultant through Ramboll.

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
