# Peer review of "Evaluating NOx emissions and their effect on O3 production in Texas using TROPOMI NO2 and HCHO"

_Atmospheric Chemistry and Physics, 2022_

## Author Comment (AC1)

**Reviewer 1:**

This manuscript presents a comprehensive study of NOx emissions and ozone chemistry in east Texas in the warm season of 2019. It contains CAMx chemical transport model (CTM) simulations, comparisons of TROPOMI NO2 retrievals with different retrieval versions and averaging kernel/AMF applications, NOx emission estimates using exponentially modified Gaussian (EMG) function method and the flux divergence method, and satellite vs. model HCHO:NO2 ratio comparison.

Thank you for the careful review. Please see our responses below in red. All new text added to the manuscript is described in detail in this document and has been *italicized*.

I have some concerns about the NOx:NO2 ratio and the NO2 effective lifetime used in the EMG- and flux divergence-based NOx emission estimates. The EMG method has been out there for over a decade, and the original assumptions about NOx:NO2 and NO2 effective lifetime seem too simplistic now, especially in this study where a full CTM simulation concurrent with satellite observations is available.

Thank you for the suggestion to further probe the NOx/NO2 ratio. We have now done so using our model simulation, and included this information as Appendix C. We calculate NOx/NO2 ratios for the partial columns below 2 km and show a spatial image of the quantity at 1 PM – the TROPOMI overpass time. The NOx/NO2 ratio above 2 km is inappropriate for use in the EMG method since the column above 2 km represents "background conditions" and is subtracted out when using the EMG method. We've now added a discussion in Appendix C on this topic:

*"To further investigate whether the $NO_X/NO_2$ ratio used in our study is appropriate, we probe the CAMx simulation to calculate the $NO_X/NO_2$ ratio for the partial columns below 2 km. The $NO_X/NO_2$ ratio above 2 km is inappropriate for use in the EMG method since the column above 2 km represents background conditions and is subtracted out when using the EMG method.*

*The $NO_X/NO_2$ ratio for the partial column below 2 km in urban areas is 1.31 ± 0.02 (Dallas: 1.33, Austin 1.30, San Antonio: 1.32, Houston: 1.295). For urban areas, this represents an uncertainty of less than 10% in the $NO_X/NO_2$ ratio. Our original assumption of using a $NO_X/NO_2$ ratio of 1.32 is warranted.*

Therefore, the assumption of a NOx/NO2 ratio of 1.32 in urban areas is supported.

However, the $NO_X/NO_2$ ratio can vary more substantially near large point sources, and we've now included a discussion of this in Appendix C:

*"However, the $NO_X/NO_2$ ratio can vary more substantially near large point sources. In the grid cells of the large point source itself, the $NO_X/NO_2$ ratio can be as large as 1.52. It is possible that the $NO_X/NO_2$ ratio in the model may be underestimated due to the emissions being equally spread out across the 4 km grid cell. $NO_X/NO_2$ ratios can be as large as 2 within 100 m downwind of major $NO_X$ sources, especially under*

*low ozone (< 30 ppb) conditions (Kimbrough et al., 2017). However, further downwind (>4 km) of these large point sources, the $NO_X/NO_2$ ratio quickly converges back to a value of ~1.31."*

The NO2 effective lifetime seems curiously low, making the EMG-based NO2 lifetime all over the place from the literature (from 6.7 h in Valin et al. 2013 to ~1 h in this and recent studies). Given the seesaw relationship between effective lifetime and emission rate, this also gives significant leverage to the emission estimates. I hope there could be some discussion and preferably quantitative assessment of the effective lifetime using the modeled NO2 reaction/removal rates.

We acknowledge the "seesaw relationship" between the $NO_2$ lifetime and $NO_X$ emissions, and we make every effort to ensure that the EMG method is outputting an effective $NO_2$ lifetime that is in the realm of possibilities. When we visually inspect and see a NO2 plume dissipating rapidly, we expect the EMG fit to output a short effective $NO_2$ lifetime; in this instance a long $NO_2$ lifetime (~6 h) would not be appropriate. We've added the following text to Section 4.1:

*"Most top-down methods fit both the effective $NO_2$ lifetime and $NO_X$ emissions simultaneously and therefore have a "seesaw relationship" – as lifetime increases, $NO_X$ emissions decrease given a constant $NO_2$ burden. Here, we visually inspect the plume to ensure that the $NO_2$ effective lifetime is reasonable given the plume decay before proceeding."*

When the EMG fit does not show an appropriate $NO_2$ lifetime, we do not report the results; this occurred for San Antonio and Houston, and is why EMG results for these cities are not included, which we noted in the text in Section 4.1:

*"The technique was applied to other urban areas, but those cities have large point sources at the periphery of the urban areas which adversely affected the calculation of the effective $NO_2$ lifetime needed to calculate the $NO_X$ emissions."*

Doing a rigorous analysis of why the NO2 effective lifetime varies by location and by platform (model vs. satellite) is beyond the scope of this project.

The other comments are mostly minor and also listed below.

Page 1, lines 27-28: satellite measures column density, not emission.

We have updated the text in this section to clarify that the comparison is between the inventory and NOx emissions derived from TROPOMI and not TROPOMI observations themselves. New sentence is as follows: "$NO_X$ emissions *inventories*, when using locally resolved inputs, agree with *$NO_X$ emissions derived from* TROPOMI $NO_2$ version 2.3.1 to within 20% in most circumstances, with a small $NO_X$ underestimate in Dallas-Fort Worth (– 13%) and Houston (– 20%)."

de Foy et al. (2015, 10.1016/j.atmosenv.2015.05.056) showed better agreement using OMI NO2 data, and satellite NO2 have been widely used to quantify point emission rates. Given OMI has been proven to be able to do that in many other studies, it's hard for me to believe TROPOMI with much smaller pixels cannot capture the plume. Even at the plume head the optical depth of NO2 is still far from optical saturation. Could non-equilibrium NO2/NOx ratio near the stack be a reason?

Thank you for the suggestion to look at the NO2/NOx ratio. The low NOx/NO2 ratio could be contributing in a small capacity as now discussed in Appendix C. In addition to Appendix C, in the abstract we now state:

*"Secondarily, the $NO_X/NO_2$ ratio in the model [near power plants] may be underestimated due to the 4 km grid cell size."*

In Appendix D, we now show substantial evidence that the cause of the TROPOMI low bias near power plants is related to TROPOMI not capturing the NOx plumes on a day-to-day basis. TROPOMI is having difficulty distinguishing NO2 attributed to power plants from the background NO2 concentrations especially in areas with atmospheric conditions that cause short NO2 lifetimes (such as Texas). The daily images at the Martin Lake Power Plant, TX is shown below and in Appendix D. During roughly half of cloud-free days in July 2019 can a NO2 plume been easily observed from this power plant. In contrast, for the Colstrip Power Plant in Montana – a power plant with similar magnitude of NOx emissions – a plume can be easily discerned on all cloud-free days.

**Martin Lake, TX**

[Figure]

**Colstrip, MT**

[Figure]

We do also want to point out that there have been significant declines in NOx emissions from the power sector since the de Foy et al. (2015) study, which analyzed emissions in the 2005 – 2011 timeframe. NOx emissions from the power sector (https://ampd.epa.gov/ampd/) have declined by ~76% between 2005 (3.63 million tons) and 2019 (0.86 million tons). It is feasible that if TROPOMI were observing the same power plant 17 years ago, we might not have the same issue.

We have added the following text in Section 3.4:

"The reason for the substantial disagreement in Texas is still unknown, but we do not believe this repudiates the prior evaluation for urban areas. *$NO_X$ emissions from the power sector in the U.S. have declined by 76% between 2005 (3.63 million tons) and 2019 (0.86 million tons) (https://ampd.epa.gov/ampd/). At these lower emission rates, it appears that TROPOMI is having difficulty distinguishing $NO_2$ attributed to power plants from the background $NO_2$ concentrations especially in areas, such as Texas, with atmospheric conditions that cause short $NO_2$ lifetimes – rapid plume dilution, high oxidation capacity due to large amounts of VOCs and water vapor, and high solar elevation angles. Secondarily, the $NO_X/NO_2$ ratio in the model may be underestimated due to the 4 km grid cell size (Appendix C).* The two power plants in New Mexico and Montana are located in areas with *smaller background $NO_2$*, lighter wind speeds, less VOCs and water vapor, and higher elevations; all of these factors cause the satellite sensor to be more sensitive to the $NO_X$ emissions. *TROPOMI does not have*

*the same difficulty over urban areas because the larger aggregated NO$_X$ emissions are more easily distinguishable from background concentrations. Please see Appendix D for a discussion on this topic.*

Page 2, line 7: the photolysis rate constant for NO2 is around 1e-2 s-1 for a wide range of SZA, indicating an NO2 "photochemical lifetime" of only ~100 s. I think what was meant here was "chemical lifetime" that is dominated by NO2->NOz and usually a few hours.

Thank you for catching this: "photochemical lifetime" has been modified to "chemical lifetime"

Page 3, lines 3-5: WRF-Chem should be included as an example, because studies cited below used it.

WRF-Chem is now referenced in this section.

Page 4, line 8: "and" should be "which" in "… and is also used for …". Also please elaborate how data assimilation was done from GFS to WRF.

As suggested, "and" is changed to "which"

We've now replaced "data assimilation" with "analysis nudging on the 36 km and 12 km domains." The new sentence is as follows:

"We use the 0.25° × 0.25° Global Forecasting System data assimilation system as initial conditions for the WRF meteorological model, and is also used for boundary conditions and *analysis nudging on the 36 km and 12 km domains*."

Page 5, line 5: please clarify if the power plant emissions data were only used for comparison or also used in the CAMx model simulation.

Power plant emissions data were used in both: all of the comparisons and the model simulation. This is now clarified in the text: "We included hourly-specific power plant emissions using measurements from the EPA's Clean Air Markets Division (CAMD) (https://www.epa.gov/airmarkets) as inputs into the model simulation."

Page 5, lines 17-18: it seems inadequate to only mention LNOx briefly here, given a subsection is devoted to it later. Please provide more information such as how the timing/locations of lightning were incorporated in the model and how LNOx generation is parameterized.

We've now included the following text to add to the description: ""*The LNOx processor estimates hourly, grid column-specific lightning flash rates (Luo et al., 2017; Price and Rind, 1992) using cloud top heights and Convective Available Potential Energy (CAPE) diagnosed from WRF temperature and moisture profiles. The processor then determines the ratio of intracloud lightning (IC) to cloud-to-ground lightning*

*(CG) according to the approach of Price and Rind (1993), NO yield per flash estimated by Pickering et al. (2017), and vertical distribution of resulting NO emission rates following DeCaria et al. (2005)."*

Page 5, line 24: at least the SWIR should be separated with other bands as it is a separate spectrometer, i.e., UV-VIS-NIR and SWIR.

This has been modified. And the chemicals observed in each region are now also documented: "TROPOMI measures total column amounts of several trace gases in the *Ultraviolet-Visible-Near Infrared (UV-VIS-NIR) (e.g., $NO_2$ and HCHO) and Shortwave Infrared (SWIR) (e.g., CO)* spectral regions.

Page 6, line 8: it is TM5-MP model but TM5 in page 13, line 23, which I assume refers to the same model.

Thank you for catching this: TM5 has been modified to TM5-MP on page 13.

Page 6: TROPOMI HCHO level 2 product should be described similarly to NO2, maybe more briefly.

We have now divided Section 2.2 into three subsections. Section 2.2.2 discusses the HCHO Level 2 product as follows:

*"HCHO slant column densities are derived from radiance measurements in the 328 – 359 nm spectral window of the UV-VIS-NIR spectrometer. In a similar manner to $NO_2$, HCHO is measured as a slant column – representing the amount of HCHO between the surface and detector – and is converted from a slant column to a vertical column using an AMF and information from TM5-MP. However, in contrast to $NO_2$, HCHO is reported only as a tropospheric vertical column amount since the stratospheric portion is negligible.*

*For our analysis, we use the v1.1.6 off-line (OFFL) algorithm, which was operational during the April through September 2019 timeframe. At the time of this study, there has not been a public release of TROPOMI HCHO data using the version 2 algorithm predating July 13, 2020."*

Page 6, line 34: the gridded TROPOMI NO2 averaging kernel should be described in more detail. From later mentioning it seems to be from gridded TM model outputs, which are not part of the TROPOMI product.

We have now divided Section 2.2 into three subsections. Section 2.2.3 discusses the averaging kernel. We added one sentence to describe the details of the vertical re-gridding of the averaging kernel.

"To apply the averaging kernel to the model simulation, *we first interpolate the averaging kernel from the TM5-MP vertical pressure levels to the CAMx vertical pressure levels at each horizontal grid location using linear interpolation. Once the averaging kernel is on the CAMx grid,* we multiply..."

Page 7, line 14: please justify the usage of 100-m wind speed. The wind speed directly scales with effective lifetime and emission rate according to equation 2. By using different wind speed, one may practically scale the emission rate estimates.

Thank you for this suggestion and opportunity to clarify. You are correct in that any errors in the wind speed will scale linearly with emission rate estimates. We have now added a section, Appendix B to address this:

"To justify, the use of the ERA5 100-m winds (as opposed to another vertical level or interval), we use the $NO_2$ column information from CAMx to determine the weighted column mid-point. Using the shape profiles described in Figure 3, we find that 50% of the tropospheric $NO_2$ column in the Dallas - Fort Worth area is below 227 m in altitude (and therefore 50% is above this); this is the weighted column mid-point. Using the WRF simulation, we find that the 100-m wind speed is 6% slower than the 227 m wind speed in Dallas – Fort Worth. However, as we discuss in Section 4.1, errors due to wind meandering (~25%) are far more critical.

We can then apply uncertainty bounds to this. In the most polluted sections of the city, the column mid-point would be lower (10s of m), and in the least polluted sections of the city the column mid-point can be as high as 500 m. While neither of these are appropriate for an areawide average, they can constrain the uncertainties of the column midpoint. Using the WRF simulation, we find that winds at 500 m are 15% larger and surface winds are 24% lower than the 100 m wind speed."

**Table B1. Wind speeds at Dallas – Fort Worth for the April – Sept 2019 average at various vertical levels in comparison to the 100-m wind speed**

| Wind fields | Ratio to 100-m wind speed |
|---|---|
| 10-m winds | 0.76 |
| 100-m winds | 1 |
| 227-m winds | 1.06 |
| 500-m winds | 1.15 |

For these reasons, in Section 4.1, we applied a broad 25% uncertainty to the wind speed and direction.

Page 7, line 21: Phi should be a "Gaussian" CDF, not any generic CDF.

Thank you for catching this: "cumulative distribution function" has been modified to "*Gaussian* cumulative distribution function".

Page 7, line 27: the NOx:NO2 ratio of 1.32 has been widely used in EMG studies and alike. Later a 10% uncertainty was assigned to it (page 18, line 27). The Beirle et al. (2021) paper cited here actually used ozone mixing ratio for a more accurate estimation. The 1.32 number came from Seinfeld and Pandis that explicitly used [NOx] = 100 ppb, [O3] = 100 ppb, and jNO2 = 0.015 s-1. Given this is a CTM study, it seems hardly justified to still use such a simplistic treatment.

Thank you for the suggestion to further probe the NOx/NO2 ratio. We have now done so using our model simulation, and included this information as Appendix C. We calculate NOx/NO2 ratios for the partial columns below 2 km and show a spatial image of the quantity at 1 PM – the TROPOMI overpass time. The NOx/NO2 ratio above 2 km is inappropriate for use in the EMG method since the column above 2 km represents "background conditions" and is subtracted out when using the EMG method. We've now added a discussion in Appendix C on this topic:

*"To further investigate whether the $NO_X/NO_2$ ratio used in our study is appropriate, we probe the CAMx simulation to calculate the $NO_X/NO_2$ ratio for the partial columns below 2 km. The $NO_X/NO_2$ ratio above 2 km is inappropriate for use in the EMG method since the column above 2 km represents background conditions and is subtracted out when using the EMG method.*

*The $NO_X/NO_2$ ratio for the partial column below 2 km in urban areas is 1.31 ± 0.02 (Dallas: 1.33, Austin 1.30, San Antonio: 1.32, Houston: 1.295). For urban areas, this represents an uncertainty of less than 10% in the $NO_X/NO_2$ ratio. Our original assumption of using a $NO_X/NO_2$ ratio of 1.32 is warranted.*

Therefore, the assumption of a NOx/NO2 ratio of 1.32 in urban areas is supported.

Equations 2 and 3: please clearly distinguish the two "emissions". One is emission rate in amount of substance per unit time, and the other one is (emission) flux in amount of substance per unit time per unit area.

This is now clarified in the text: "*Equation 2 yields emission estimates in units of mol-s$^{-1}$*" and "*The divergence of the fluxes yields an emission estimate in units of mol-m$^{-2}$ s$^{-1}$. The fluxes can then be multiplied by the urban area to get emission rates in analogous units as Equation 2*.

Page 8, line 11: why were only the central 250 across track positions used?

The central 250 pixels have a higher resolution and more vertical column than the outer bands, and therefore give a better spatial resolution of the sources. We have modified the text as follows:

"The central 250 pixels (out of 450) were used *as these have a higher resolution than the outer bands. We retrieved swaths* from October 2019 through September 2021."

Page 8, line 16-17: the sentence about lifetime scales with the length scale and equals to L/(2U) does not seem to be connected with the context. Please check if it was supposed to be there.

Thank you for pointing this out and our apologies for our mistake. We did not update the text from our earlier procedure. The correct description is described in the text above, which remains unchanged:

"Sinks of NO2 are included in the equation by adding VCD divided by the atmospheric lifetime of NO2, τ, which was taken from the EMG fit."

We can indeed remove the last 2 sentences which no longer apply:

"The lifetime is linearly dependent on the length scale and inversely proportional to the wind speed (tau = L / 2U). For the length scale we use the geometric mean of the radii of the Gaussian ellipses, which were calculated using the covariance matrix."

Figure 3: please explain what the red solid dots and the black dashed lines mean. The caption states on the right there are NO2 shape profiles from "two model simulations", but I can only see one profile in each panel.

Thank you for catching this: this figure has been updated, and the figure caption has been clarified as requested. The profiles on the right panels were overlapping due to the thickness of the line. The lines are now thinner.

Page 12, line 14: "model out" should probably be "model output".

Thank you for catching this: "model out" has been modified to "model *output*".

Figure 4: in the third panel, dividing the tropospheric column into "below 2km", "lightning NOx", and "other" seems a bit strange, as it mixed up vertical layers with sources. An implicit assumption here is that LNOx is exclusively constrained from 2 km to tropopause. Although this could be a good assumption, it is better to add some clarification in the text.

Figure 3 demonstrates that the inclusion of lightning NOx emissions only meaningfully affects $NO_2$ concentrations above 2 km. In section 3.2 we add the sentence: *"The inclusion of lightning $NO_X$ emissions only meaningfully affects $NO_2$ concentrations above 2 km in altitude."*

Page 13, lines 8-10: "while the urban NO2 will artificially decrease" appeared twice.

Thank you for catching this: the first instance, including the entire sentence, has been removed.

Page 13, lines 17-20: these hypotheses also appeared in the abstract and in page 16, lines 7-10 but in slightly different forms. Please consider consolidating and reconciling them. It also a bit strange as hypotheses were formed but not followed by testing or design of testing.

We have removed this hypothesis in this location and reframed in Section 3.4 and Appendix D. The full text is as follows:

"The reason for the substantial disagreement in Texas is still unknown, but we do not believe this repudiates the prior evaluation for urban areas. *$NO_X$ emissions from the power sector in the U.S. have declined by 76% between 2005 (3.63 million tons) and 2019 (0.86 million tons) (https://ampd.epa.gov/ampd/). At these lower emission rates, it appears that TROPOMI is having difficulty distinguishing $NO_2$ attributed to power plants from the background $NO_2$ concentrations especially in areas, such as Texas, with atmospheric conditions that cause short $NO_2$ lifetimes – rapid plume dilution, high oxidation capacity due to large amounts of VOCs and water vapor, and high solar elevation angles. Secondarily, the $NO_X/NO_2$ ratio in the model may be underestimated due to the 4 km grid cell size (Appendix C).* The two power plants in New Mexico and Montana are located in areas with *smaller background $NO_2$*, lighter wind speeds, less VOCs and water vapor, and higher elevations; all of these factors cause the satellite sensor to be more sensitive to the $NO_X$ emissions. *TROPOMI does not have the same difficulty over urban areas because the larger aggregated $NO_X$ emissions are more easily distinguishable from background concentrations. Please see Appendix D for a discussion on this topic.* Future work should focus on evaluating the $NO_2$ from power plants, such as *in situ* measurements from aircraft and ground-based vertical column instruments (e.g., Pandora (Herman et al., 2009)).

Page 15, line 6: as correlation is mentioned in the text, the associated number should be correlation coefficient, r, not r2, which appears to be coefficient of determination. It is denoted as R2 in Appendix A though.

We've modified the word "correlation" to the phrase "*association with each other*". The value displayed on the image is indeed the $r^2$.

Figure 6: the caption seems incorrect as "bias-corrected" and "downscaled" TROPOMI NO2 was never mentioned.

Thank you for catching this. New figure caption is as follows: "$NO_2$ tropospheric vertical column amounts averaged across April through September 2019 from TROPOMI, *TROPOMI v2.3.1, TROPOMI v2.3.1 with new AMF*, and CAMx for the largest four cities (Dallas, San Antonio, Austin and Houston)."

Page 16, line 10: high solar zenith angles sounds the opposite. Should it be either low SZA or high solar elevation angle?

Thank you for this suggestion. We modified "high solar zenith angles" to "high solar *elevation* angles"

Page 16, lines 7-10: first, it is questionable how much the "effective lifetime" fitted from EMG can represent real atmospheric chemistry. The 0.5 hour fitted from Fig. 8 right look especially unrealistic. Since this study presents a full CTM simulation, could the reaction rates in the model shed some light on the reliability of the "effective lifetime"?

We also agree that a 0.5 hour effective lifetime seems rather short, but investigating the reasons behind this is beyond the scope of this manuscript. As discussed in Section 4.1, we feel that a poor representation of vertical transport may be the primary cause of this and not a simplified representation of the chemistry.

Page 16, lines 7-10: second, TROPOMI only measures column amount, the factors given (wind, OH, VOC, high sun) do not seem to matter how well TROPOMI measures NO2 column amount.

We had been referring to TROPOMI's ability to distinguish $NO_2$ pulses in a location with short $NO_2$ chemical lifetimes. Because the chemical lifetime of $NO_2$ is shorter in Texas than more northerly latitudes, the $NO_2$ concentrations are therefore lower, and harder to detect by TROPOMI. This section has now been re-phrased, we now state: "*$NO_X$ emissions from the power sector have declined by 76% between 2005 (3.63 million tons) and 2019 (0.86 million tons). At these lower emission rates, it appears that TROPOMI is having difficulty distinguishing $NO_2$ attributed to power plants from the background $NO_2$ concentrations especially in areas, such as Texas, with atmospheric conditions that cause short $NO_2$ lifetimes – rapid plume dilution, high oxidation capacity due to large amounts of VOCs and water vapor, and high solar elevation angles.*"

Page 18, lines 14-20: it would be useful to clarify photochemical and chemical lifetimes of NO2 here, and how the effective lifetime from EMG is related to them.

This is discussed in de Foy et al. 2014, and we have briefly summarized here: "*The effective lifetime is a function of the chemical lifetime and dispersion lifetime*:

$$\frac{1}{\tau_{effective}} = \frac{1}{\tau_{chemical}} + \frac{1}{\tau_{dispersion}} \qquad (4)$$

The *effective lifetime* could be *increased* in a model simulation by increasing the $NO_2$ chemical lifetime (e.g. slower photolysis, slowing the $NO_2$+OH reaction rate, faster recycling of $NO_Z$ ($NO_Z$ = Alkyl nitrates, PAN, and $HNO_3$) back to $NO_2$) or *by increasing vertical mixing (less plume meandering at higher altitudes due to less surface frictional interactions).*

Page 18, line 23: DISCOVER-AQ in Texas in September 2013 provided lots of vertically resolved NOx measurements already.

This is true, however, we only felt comfortable comparing a 2019 simulation to 2013 in the free troposphere (above 3000 m / 10000 ft) since there is less interannual variability in this vertical domain. DISCOVER-AQ was an especially fruitful campaign, but almost no measurements were acquired above 10,000 ft in altitude.

In page 10 and 18, we clarify to demonstrate our request for measurements in the free troposphere: "Collocated $NO_2$ measurements *in the free troposphere* in time and space would be needed to evaluate this further."

Page 19, line 13: please provide the power plants NOx are biased low by how much.

Thank you for catching this, we've now provided the quantity of 65% in this sentence: "The results from the flux divergence method are consistent with the results from the EMG method (i.e., Dallas $NO_X$ is within 20% and power plants $NO_X$ are *biased low by 65%*) provided that a short $NO_2$ lifetime is assumed."

Figure 11: please explain what the numbers in the plot mean.

The numbers are the ratio for each location. We've now clarified on the figure caption: *"The ratios from the satellite and model for each area are shown directly on the plot."*

---

## Author Comment (AC2)

**Reviewer 2:**

This study evaluates NOx emissions and ozone production in Texas, which covers several interesting questions, including model-satellite comparison, top-down NOx emission estimates using EMG approach, and HCHO/NO2 for identifying the ozone production regimes. Overall the results are clearly presented, and the figures are informative.

Thank you for the careful review. Please see our responses below in red. All new text added to the manuscript is described in detail in this document and has been *italicized*.

There are several interesting findings, which could well be three independent studies, but my concern is that the findings are a little disconnected, and it is not easy to tell the major take-aways from the study. I'd suggest the authors better frame the core questions, and show how each section is connected.

We have shortened the Abstract and made it clearer. We have also re-framed sections of the Conclusions to better describe the take-away points.

Below are some major comments:

1. It is not clear to me why the authors compare two different versions of TROPOMI NO2. Satellite retrieval products are updated routinely. The newest version should certainly be better, and the older version is already replaced with newer one. I don't think there is much scientific value of such comparison. Evaluation of different versions should be in the technical document of TROPOMI NO2. I'd suggest the authors stick to the newest, publicly available version.

In this revised draft, we have removed mentions of the Version 1.3 algorithm in Sections 3.4 (including Figures 6 & 7) and 4.1, as we agree that it was unnecessary to discuss. Please see new versions of Figures 6 & 7 below:

[Figure]

**Figure 6.** NO₂ tropospheric vertical column amounts averaged across April through September 2019 from TROPOMI, TROPOMI v2.3.1, TROPOMI v2.3.1 with new AMF, and CAMx for the largest four cities (Dallas, San Antonio, Austin and Houston). (Right) Scatterplot showing slope and correlation of various TROPOMI configurations and CAMx

[Figure]

**Figure 7.** NO₂ tropospheric vertical column amounts averaged across April through September 2019 from TROPOMI v2.3.1, TROPOMI v2.3.1 with new AMF, and CAMx for the largest three power plants in East Texas (Martin Lake [Lat: 32.25 ° N, Lon: 94.58° W], Limestone [Lat: 31.42º N, Lon: 96.25º W], and Sam Seymour [Lat: 29.92º N, Lon: 96.75º W]). (Right) Scatterplot showing slope and correlation of various TROPOMI configurations and CAMx

However, we decided to keep the version 1.3 algorithm versus version 2.3.1 comparison in Section 3.1 because there is value in doing a US-specific comparison between algorithms which is not covered in either technical document by developers. Inclusion of such a comparison is important because many users are most familiar with prior versions of the algorithms and have been wondering how the algorithms compare on a regional basis, rather than a global basis. We show evidence that regionally, the changes match global changes; a simple but important confirmation.

2. Related to previous question, I'd suggest the authors use newest version TROPOMI HCHO and NO2 data in Section 4.2 to evaluate ozone sensitivity. The authors have shown better performance of v2, but switch to v1 in 4.2. It is also not clear to me why the authors use an outdated version of TROPOMI HCHO (v1.1) while a newer version is already available.

A new HCHO product is not available for the timeframe in which we are investigating. The Version 2 algorithm only has data between July 13, 2020 and present, and not during 2019.

3. It's also not clear to me why satellite HCHO is less sensitive to the application of AK. While it may not help improve the spatial patterns of HCHO, application of AK may resolve the overall difference between modeled and satellite HCHO. It'd be great if authors can show a figure or two to illustrate this point.

We have now added an additional panel on to Figure 10 to show the effect of the averaging kernel. The averaging kernel affects the column HCHO amounts by ±2.5% for areawide averages. We have now

*added the following text: "The difference between CAMx and CAMx with the averaging kernel applied is ±2.5% for areawide averages."*

[Figure]

Specific comments:

1. Figure 3 (right): What does the red line mean? Which shape profile?

The red line is the shape profile. It represents how much of the partial column is at any given altitude. The shape of the line is equivalent to the $NO_2$ mixing ratio show in the left panel, but normalized to a unitless quantity that integrates to unity over the depth of the troposphere.  The profile on the right looks different because it is on a linear scale while the left panel is on a log-scale. We have now added the following clarifying text:

"In the right panels of Figure 3, we show *the modeled shape profiles – the $NO_2$ vertical distribution normalized to a unitless quantity that integrates to unity over the depth of the troposphere…"*

2. Figure 4: There is no explanation of the right figure in the caption. Why is it continuous? Are these generated for selected pixels?

Thank you for catching this oversight. We have now clarified the right hand panel with the following text:

*"The fraction of the $NO_2$ column attributed to different layers of the atmosphere (below 2 km, above 2 km (attributed to Other), and above 2 km attributed to lightning $NO_X$ (LNOx)) at six locations (Gulf of Mexico, rural Central Texas, Austin, San Antonio, Dallas and Houston); the fraction attributed to lightning $NO_X$ (LNOx) is calculated as the $NO_2$ addition between the two simulations without and with lightning $NO_X$ emissions."*

3. Page 16 Lines 5 to 10: The authors show TROPOMI NO2 is significantly lower than modeled NO2, and they attribute this to issues with TROPOMI. While it may be true, but could this also be due to model unable to capture the sub-grid processes?

Thank you for the suggestion to further probe sub-grid processes, and the one most relevant here is the NOx/NO2 ratio. We have now probed the NOx/NO2 ratio using our model simulation, and included this information as Appendix C. We calculate NOx/NO2 ratios for the partial columns below 2 km and show a spatial image of the quantity at 1 PM – the TROPOMI overpass time. The NOx/NO2 ratio above 2 km is inappropriate for use in the EMG method since the column above 2 km represents "background conditions" and is subtracted out when using the EMG method. We've now added a discussion in Appendix C on this topic:

"the $NO_x/NO_2$ ratio can vary more substantially near large point sources. In the grid cells of the large point source itself, the $NO_x/NO_2$ ratio can be as large as 1.52. It is possible that the $NO_x/NO_2$ ratio in the model may be underestimated due to the emissions being equally spread out across the 4 km grid cell. $NO_x/NO_2$ ratios can be as large as 2 within 100 m downwind of major $NO_x$ sources, especially under low ozone (< 30 ppb) conditions (Kimbrough et al., 2017). However, further downwind (>4 km) of these large point sources, the $NO_x/NO_2$ ratio quickly converges back to a value of ~1.31."

In addition to Appendix C, in the abstract we now state:

"Secondarily, the $NO_x/NO_2$ ratio in the model [near power plants] may be underestimated due to the 4 km grid cell size."

With that said, in Appendix D, we now show substantial evidence that the cause of the TROPOMI low bias near power plants is related to TROPOMI not capturing the NOx plumes on a day-to-day basis. TROPOMI is having difficulty distinguishing NO2 attributed to power plants from the background NO2 concentrations especially in areas with atmospheric conditions that cause short NO2 lifetimes (such as Texas). The daily images at the Martin Lake Power Plant, TX is shown below and in Appendix D. During roughly half of cloud-free days in July 2019 can a NO2 plume been easily observed from this power plant. In contrast, for the Colstrip Power Plant in Montana – a power plant with similar magnitude of NOx emissions – a plume can be easily discerned on all cloud-free days.

**Martin Lake, TX**

[Figure]

[Figure]

**Colstrip, MT**

[Figure]

We do also want to point out that there have been significant declines in NOx emissions from the power sector since the de Foy et al. (2015) study, which analyzed emissions in the 2005 – 2011 timeframe. NOx emissions from the power sector (https://ampd.epa.gov/ampd/) have declined by ~76% between 2005 (3.63 million tons) and 2019 (0.86 million tons). It is feasible that if TROPOMI were observing the same power plant 17 years ago, we might not have the same issue.

We have added the following text in Section 3.4:

"The reason for the substantial disagreement in Texas is still unknown, but we do not believe this repudiates the prior evaluation for urban areas. *NO$_X$ emissions from the power sector in the U.S. have declined by 76% between 2005 (3.63 million tons) and 2019 (0.86 million tons) (https://ampd.epa.gov/ampd/). At these lower emission rates, it appears that TROPOMI is having difficulty distinguishing NO$_2$ attributed to power plants from the background NO$_2$ concentrations especially in areas, such as Texas, with atmospheric conditions that cause short NO$_2$ lifetimes – rapid plume dilution, high oxidation capacity due to large amounts of VOCs and water vapor, and high solar elevation angles. Secondarily, the NO$_X$/NO$_2$ ratio in the model may be underestimated due to the 4 km grid cell size (Appendix C).* The two power plants in New Mexico and Montana are located in areas with *smaller background NO$_2$*, lighter wind speeds, less VOCs and water vapor, and higher elevations; all of these factors cause the satellite sensor to be more sensitive to the NO$_X$ emissions. *TROPOMI does not have*

*the same difficulty over urban areas because the larger aggregated NO$_X$ emissions are more easily distinguishable from background concentrations. Please see Appendix D for a discussion on this topic.*

4. Page 20 Line 12: This sentence is confusing.

We have modified to: "*We first compare column HCHO comparison between CAMx and TROPOMI.*"

5. Table 3 shows the derived NOx emissions are sensitive to NO2 lifetime. How much confidence do you have with the NOx lifetime in EMG approach? Does it agree with the NO2 lifetime simulated from model? And how does the seasonal variation in NO2 lifetime affect the calculation of emissions?

We acknowledge the "seesaw relationship" between the NO$_2$ lifetime and NO$_X$ emissions, and we make every effort to ensure that the EMG method is outputting an effective NO$_2$ lifetime that is in the realm of possibilities. When we visually inspect and see a NO2 plume dissipating rapidly, we expect the EMG fit to output a short effective NO$_2$ lifetime; in this instance a long NO$_2$ lifetime (~6 h) would not be appropriate. We've added the following text to Section 4.1:

*"Most top-down methods fit both the effective NO$_2$ lifetime and NO$_X$ emissions simultaneously and therefore have a "seesaw relationship" (Liu et al., 2022)– as lifetime increases, NO$_X$ emissions decrease given a constant NO$_2$ burden. Here, we visually inspect the plume to ensure that the NO$_2$ effective lifetime is reasonable given the plume decay before proceeding."*

When the EMG fit does not show an appropriate NO$_2$ lifetime, we do not report the results; this occurred for San Antonio and Houston, and is why EMG results for these cities are not included, which we noted in the text in Section 4.1: "The technique was applied to other urban areas, but those cities have large point sources at the periphery of the urban areas which adversely affected the calculation of the effective NO$_2$ lifetime needed to calculate the NO$_X$ emissions."

Doing a rigorous analysis of why the NO2 effective lifetime varies by location and by platform (model vs. satellite) is beyond the scope of this project.

Through unpublished work (that is beyond the scope of this project), we have found that during winter, the NO$_2$ dispersion lifetime is dominant, and therefore makes this method much more difficult to apply. In this study, we only focus on the warm season months (April – Sept) when the chemical lifetime is shorter.

6. Figure 12: It's interesting to see there is almost no diurnal cycle with HCHO. I think both biogenic emissions and photolysis rate vary diurnally. It'd be interesting to illustrate why HCHO is relatively constant, and what this would mean for ozone production.

We agree that this is interesting, but an investigation of the diurnal patterns of HCHO is beyond the scope of this project, particularly because this would involve vertically-resolved HCHO measurements (which are

sparse) and disentangling the complex relationship between HCHO and VOC emissions. We refer you (and the reader) to Schroeder et al. (2016) and Schwantes et al., (2022), which we now cite in this location:

*"HCHO has broad peak in the afternoon, which is likely related to biogenic emissions and secondary formation. However, the HCHO diurnal cycle is flatter than we expected; this may be due to model difficulties in representing complex VOC chemistry for secondary HCHO production (Schroeder et al., 2016; Schwantes et al., 2022)."*

7. Figure A1: There seems to be a large discrepancy between CAMx and observed NOx and NOy. I'd suggest the authors investigate why CAMx is biased too low, and how this would affect the interpretation of other findings.

TCEQ $NO_2$ ground monitors are disproportionately located near or directly adjacent to major highways; often within 1 km but sometimes within 50 m. While this is appropriate from a regulatory standpoint, it is often inappropriate to compare these monitors to a model simulation with 4 x 4 $km^2$ resolution. Therefore, a low bias in the model as compared to the monitors is primarily indicative of the sub-grid processes that CAMx cannot resolve. This is why a satellite comparison to the model is more appropriate than a ground monitor comparison. We now cite Souri et al. 2022 which discusses how care is needed when comparing pointwise measurements to concentrations spatially averaged over larger (>1 km) grid cells:

*"As discussed in (Souri et al., 2022), care is needed when comparing pointwise measurements to concentrations spatially averaged over large (>1km) grid cells."*

---

## Author Response (AR2)

Reviewer #1

The authors made good efforts to address the referees' comments. Here are a few more minor comments.

Thank you for your careful re-review. The additional text added has been *italicized*.

The revision has "Here, we visually inspect the plume to ensure that the NO2 effective lifetime is reasonable given the plume decay before proceeding", and the response has "When the EMG fit does not show an appropriate NO2 lifetime, we do not report the results". I suggest make it less subjective by quantifying boundaries, i.e., how long is considered "reasonable" or "appropriate".

Modified to, "Here, we visually inspect the plume to ensure that the $NO_2$ effective lifetime is reasonable *(generally between 0.5 – 5 hours)* given the plume decay before proceeding."

"Section 2.2.2 of the revised manuscript, slant column observed by a satellite should be the amount from the sun to the detector, not from surface to the detector. That part can be just removed.

The phrase, "representing the amount of HCHO between the surface and detector" was removed as suggested.

"Page 8, line 24, "The fluxes can then be multiplied by the urban area to get emission rates". Strictly, this should be a 2D integration.

Modified to, "The fluxes can then be *integrated across* the *2-D* urban area to get emission rates in analogous units as Equation 2."

Page 8, line 30, "The central 250 pixels (out of 450)". The same argument can be applied to NO2, but all NO2 across-track positions are used. Is it more critical to only use the swath center for HCHO? Is there a reference for such filtering?

We found it to be critical for this particular method as we currently state in the text. It wasn't critical for the other uses of the TROPOMI NO2 data.

Page 16, line 15 of the revised manuscript, "evaluating the NO2 from power plants and the NOX/NO2 ratio as the plume involves". Should "involves" be "evolves"?

Good catch. Modified from "involves" to "evolves"

Page 19, line 12-13, it may be a stretch to say EMG and flux divergence are "consistent" when one biases low by 65%. Consider rephrasing it or adding some discussion.

Thank you for pointing this out. Yes, there is still disagreement between methods at the power plant location. Modified to, "The results from the flux divergence method are consistent with the results from the EMG method *in the Dallas area* provided that a short $NO_2$ lifetime is assumed."

Reviewer #2

The authors have well addressed reviewers' comments. Great job!